# Early Quantization Shrinks Codebook:
# A Simple Fix for Diversity-Preserving Tokenization

## Abstract

While discrete tokenizers are suspected to inherently limit sample diversity in token-based generative models, we show this diversity gap is not caused by discretization itself, but rooted in the timing of quantization. In this study, we systematically identify quantization in the initial stage as the primary catalyst for a representational misalignment, where the codebook suffers a premature coverage failure, anchoring to a narrow latent space. This initial coverage deficit prevents the codebook from capturing the diverse embedding space of the encoder. Though this may yield deceptively strong reconstructions, it creates a bottleneck that forces the generator to rely on a homogenized set of tokens. Ultimately, the codebook's failure to anchor to robust representations at the onset of training impairs generative variety and limits sample diversity. To address this, we propose Deferred Quantization, a simple yet effective strategy that introduces a separate continuous learning phase. By allowing the encoder to first establish a well-distributed representation space before introducing discretization, the codebook can effectively anchor to a mature and diverse latent landscape. Across tokenizers and token-based generators, Deferred Quantization consistently mitigates this coverage failure, improves generative diversity, and preserves reconstruction and compression. We additionally provide a coverage diagnostic suite and offer practical guidance for designing diversity-preserving discrete tokenizers.

## 1 Introduction

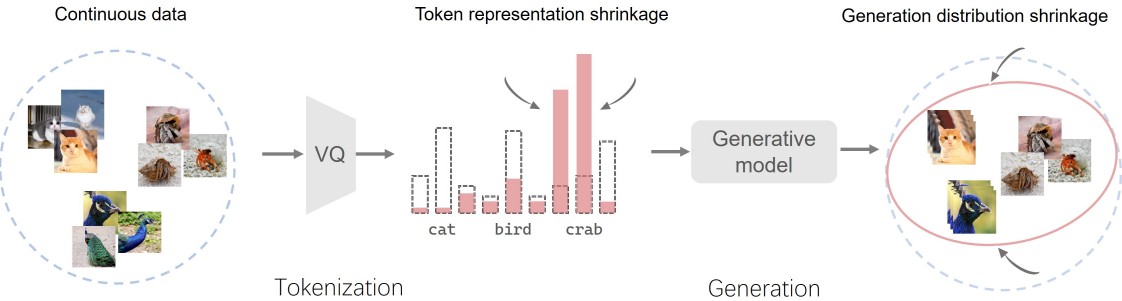

Figure 1: **Token coverage failure degrades the diversity of generative model.** Vector quantization is a widely used technique to map continuous data into discrete tokens, which assists the transformer-based generative model's generation. We observe that token coverage failure, manifested as narrow distribution in latent space, leads to a restricted distribution of the generated data.

Transformer-based generative models for autoregressive generation have gained significant popularity in recent years in the field of image generation. These models underpin many state-of-the-art systems such as DALL-E (Betker et al., 2023) and VAR (Tian et al., 2024), which have found wide-ranging applications in art creation, design automation, and data augmentation. Their practical value lies not only in producing visually compelling images but also in enabling new workflows for creative and industrial domains.

Despite their success, transformer-based generative models may exhibit a limitation: the synthetic images they produce tend to cover a narrower distribution than real images. This phenomenon, commonly referred to as mode collapse, results in limited diversity in the generated content. Mode collapse causes the model to ignore valid variations in the data distribution, which limits its generalization, realism, and utility in downstream tasks. In this work, we study the ability of generative models to produce diverse high-quality outputs from the perspective of tokenization, and study how limitations in diversity arise in token-based generation pipelines.

A common implicit assumption in token-based generation is that a tokenizer is good if it yields low reconstruction error. However, reconstruction primarily reflects performance on the high-density regions of the data distribution, while diversity depends critically on broader coverage of the representation space, including low-density regions. This mismatch suggests that a tokenizer can appear strong under reconstruction-centric evaluation, yet still bottleneck downstream diversity.

To address mode collapse and improve generative fidelity and diversity, various studies have proposed architectural innovations or alternative training objectives. For example, VQGAN (Esser et al., 2021) incorporates vector quantization to learn a discrete codebook, while ImageGPT (Chen et al., 2020a) treats images as sequences of pixels to better capture complex data distributions. However, these advances do not directly characterize whether the learned discrete tokens sufficiently cover the latent representation space required for diverse generation.

In this work, we identify a previously overlooked yet critical aspect of tokenization, which we term *token coverage failure*. It describes a phenomenon in which the learned token embeddings fail to span the full embedding space and tend to cluster into a limited region of the distribution, as illustrated in Fig. 1. When token coverage is restricted, the generated outputs are also constrained to a narrow portion of the data space, resulting in reduced diversity and diminished modality coverage. Therefore, token coverage failure provides a concrete mechanism by which tokenization can limit downstream diversity even when reconstruction appears satisfactory.

Importantly, existing studies (Zhu et al., 2025; Yu et al., 2021; 2023b; Mentzer et al., 2023) on codebook collapse have mainly focused on the dead-token problem, where some codebook entries are never or rarely used during training, while largely overlooking the issue of insufficient coverage of the embedding space, *i.e.*, token coverage failure. This distinction matters because a codebook may exhibit healthy usage statistics while still occupying only a small region of the latent space, thereby restricting the support available to downstream generative models.

To better understand this issue, we analyze its underlying cause and identify a key contributing factor: the commonly used token initialization strategy (Zeghidour et al., 2021) in VQ training. In typical practice, token embeddings are initialized from the outputs of an untrained encoder, which already form a compact and clustered distribution in the latent space. This initialization bias restricts the subsequent expansion of the token space during training, making it difficult for the learned tokens to spread out and align with the true data distribution. As a result, it induces token coverage failure.

We then introduce a simple intervention, Deferred Quantization that alleviates this coverage deficit and enables a controlled study of its impact: we first pretrain the encoder without VQ and then utilize pretrained encoder embeddings to initialize the codebook. This decouples representation learning from discretization in the early stage, allowing the encoder to learn meaningful representations before quantization is introduced. As a result, it reduces the resistance faced during VQ optimization and mitigates the coverage failure. Using this intervention, we validate the presence and impact of token coverage failure through extensive experiments on a variety of synthetic and real-world datasets. Across these settings, we observe that token coverage failure is associated with reduced diversity, whereas its mitigation reliably improves both the diversity and fidelity of generated images.

Our main contributions are summarized as follows:

- We analyze the detrimental effects of early quantization, demonstrating it restricts the codebook to a narrow, homogenized latent space, which consequently impairs generative sample diversity.

- We introduce Deferred Quantization, a simple yet effective training intervention that decouples feature learning from discretization.

- We validate this Deferred Quantization across synthetic and real-world datasets, showing that mitigating token coverage failure consistently restores the codebook's representational breadth and improves generative variety.

## 2 Related Works

Vector quantization plays an important role in data compression and signal processing under Shannon's rate–distortion theory (Gersho & Gray, 2012; Cover, 1999). Traditional approaches such as K-means clustering (McQueen, 1967) have been widely used, but they become computationally expensive when applied to high-dimensional data (Le Tan et al., 2018). To mitigate this challenge, DeepVQ (Le Tan et al., 2018) improved efficiency by mapping data to lower-dimensional latent spaces before quantization. Moreover, (Van Den Oord et al., 2017) proposed VQ-VAE which integrates VQ with variational autoencoders, using a straight-through estimator (Bengio et al., 2013) to handle discrete variables. To refine VQ methods for improved performance, variants such as Residual Quantization (Lee et al., 2022), Product Quantization (Chen et al., 2020b), and Soft Convex Quantization (Gautam et al., 2023) further enhanced representation capacity and efficiency. Recent advances incorporate attention mechanisms and transformer architectures (Vaswani et al., 2017; Yu et al., 2021) to dynamically select codebooks and capture global data dependencies. Recent works also explore per-channel codebooks (Hsu et al., 2024) and neural network variants of residual quantization (Huijben et al., 2024) to predict specialized codebooks, enhancing the model's expressive power.

VQ has been widely applied across various domains. In natural language processing, VQ facilitates sequence modeling (Łukasz Kaiser et al., 2018), enhancing tasks such as language modeling. In computer vision, VQ has significantly advanced image generation and compression techniques (Esser et al., 2021). Similarly, in audio processing, VQ techniques have captured complex temporal dependencies (Dhariwal et al., 2020). Furthermore, in multimodal applications, VQ supports the integration of different data types through shared discrete representations (Ramesh et al., 2021).

Despite these advancements, VQ methods encounter challenges that restrict their broader application, including but not limited to codebook collapse, training instability, and computational overhead. Extensive research has been conducted on solving the codebook collapse problem, where only a subset of tokens are used leading to inefficient representation usage and reduced diversity in outputs, by reducing token dimension (Yu et al., 2021), orthogonal regularization loss (Shin et al., 2023), multi-headed VQ (Mama et al., 2021), finite scalar quantization (Mentzer et al., 2023), and Lookup Free Quantization (Yu et al., 2023a). Recent methods like (Goswami et al., 2024) and (Baykal et al., 2024) also strive to enhance tokens usage efficiency.

Prior work has explored delaying or stabilizing quantization in VQ training. Most closely related, Łańcucki et al. (2020) introduce a warm-up phase that disables quantization in the initial iterations, allowing encoder outputs to stabilize. More broadly, recent works such as TokenBridge (Wang et al., 2025) and ReVQ (Zhang et al., 2025) also decouple continuous representation learning from discretization by starting from pretrained continuous VAEs. Although these methods share the high-level motivation of separating continuous representation learning from discretization, they differ from Deferred Quantization in both objective and training protocol. Specifically, TokenBridge (Wang et al., 2025) and ReVQ (Zhang et al., 2025) aim to transfer the reconstruction ability of a pretrained continuous VAE to a discrete tokenizer while freezing the encoder and decoder during discretization, whereas our method targets Token Coverage Failure induced by early quantization and jointly fine-tunes the encoder, decoder, and codebook after quantization is introduced. Another related direction improves codebook learning through stronger priors learned by large language models or unsupervised models, e.g., VQCT (Zhang et al., 2024) and VQGAN-LC (Zhu et al., 2024).

Our work is related to these approaches in spirit, but differs in focus. Our main contribution is a systematic diagnosis of why such strategies help: beyond improving code usage or reducing dead tokens, deferred quantization expands the latent support of the learned codebook. We show that this broader latent support is closely connected to downstream generative diversity.

## 3 Preliminary

### 3.1 Mode Collapse and Limited Diversity

Transformer-based generative models might exhibit a limitation: the synthetic images they produce tend to cover a narrower distribution than real images. This phenomenon, commonly referred to as *mode collapse*, results in limited diversity in the generated content. Mode collapse causes the model to ignore valid variations in the data distribution, which limits its generalization, realism, and utility in downstream tasks.

In this paper, we use *mode collapse* to refer to the reduction in diversity and modality coverage of generated outputs relative to the data distribution. In particular, mode collapse manifests when the generative model concentrates probability mass on a small subset of plausible outcomes, underrepresenting other valid modes of variation in the data. As a result, mode collapse and limited diversity undermine the effectiveness of generative models in settings that require broad coverage of the data distribution.

### 3.2 Vector Quantization

Following standard VQ-VAE (Van Den Oord et al., 2017), we adopt the formulation: an encoder $E_\theta$, a decoder $D_\theta$ and a set of tokens $\mathcal{T} = \{t_1, t_2, \ldots, t_S\}$. The token set $\mathcal{T}$ constitutes the codebook, which is employed to store the discretized representations. The encoder is responsible for mapping the raw data $X = \{x_1, x_2, \ldots, x_N\}$ to a set of continuous representations $\mathcal{Z} = E_\theta(X)$, where $\mathcal{Z} = \{z_1, z_2, \ldots, z_N\}$. And the decoder reconstructs the data $X' = D_\theta(\hat{Z})$ based on the set of discretized representations $\hat{Z}$, where $\hat{Z} = \{\hat{z}_1, \hat{z}_2, \ldots, \hat{z}_N\}$. The process of tokenizing a continuous representation $z_j$ to discrete representation $\hat{z}_j$ is as following:

$$\hat{z}_j = \arg\min_{t_k \in \mathcal{T}} \|z_j - t_k\|, \tag{1}$$

where $t_k$ is a token in token set $\mathcal{T}$ and $k$ is the index. This quantization is performed by finding the nearest token $t_k$ in $\mathcal{T}$.

The overall training objective is:

$$\mathcal{L} = \mathcal{L}_{\text{recon}} + \mathcal{L}_{\text{codebook}} + \beta\,\mathcal{L}_{\text{commit}}, \tag{2}$$

where $\text{sg}[\cdot]$ denotes the stop-gradient operator and $\beta$ is the commitment weight. The codebook loss $\mathcal{L}_{\text{codebook}} = \|\text{sg}[z_j] - \hat{z}_j\|_2^2$ encourages the codebook entries to move towards the encoder outputs, while the commitment loss $\mathcal{L}_{\text{commit}} = \|z_j - \text{sg}[\hat{z}_j]\|_2^2$ encourages the encoder outputs to stay close to the chosen codebook entries. The reconstruction loss $\mathcal{L}_{\text{recon}}$ is composed of mean squared error $\mathcal{L}_{\text{MSE}} = \frac{1}{N}\sum_{j=1}^{N}\|x_j - x'_j\|_2^2$, perceptual loss $\mathcal{L}_{\text{LPIPS}}$ (Zhang et al., 2018), and adversarial loss $\mathcal{L}_{\text{GAN}}$ (Esser et al., 2021). The specific combination of reconstruction losses depends on the experimental setting; details are provided in Section 6.

## 4 Token Coverage Failure

This section characterizes *token coverage failure* and analyzes why it can induce downstream mode collapse and reduced diversity in token-based generation pipelines.

**Definition.** Token coverage failure refers to a failure mode where learned token embeddings occupy only a small region of the encoder embedding space, causing many tokens to cluster around a few modes, as illustrated in Fig. 1. In the ideal case, the token distribution should align with and adequately cover the encoder embedding distribution.

**Synthetic evidence: coverage failure induces reconstruction collapse.** To validate the phenomenon, we conduct experiments on a synthetic dataset using VQ-VAE. Specifically, we use VQ-VAE to reconstruct the input data and compare the resulting token distribution with the original data distribution. The synthetic dataset comprises 10,000 data points uniformly sampled from 10 distinct Gaussian distributions (see Sec. 6.1 for details).

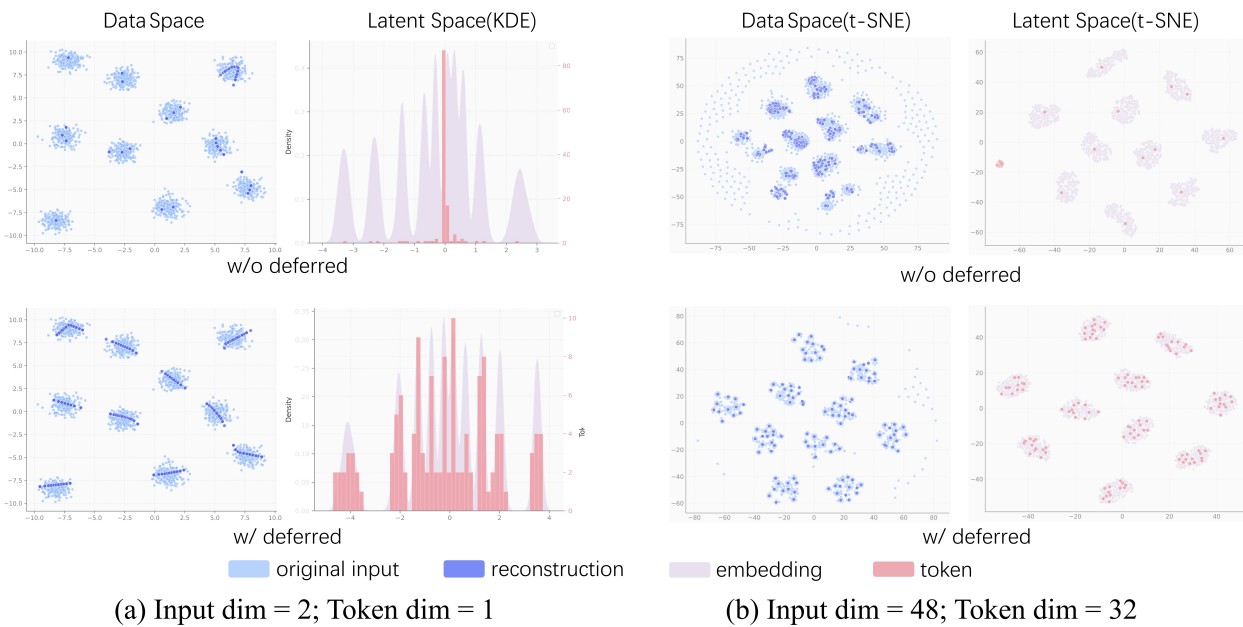

(a) Input dim = 2; Token dim = 1          (b) Input dim = 48; Token dim = 32

Figure 2: **Deferred quantization mitigates token coverage failure and reconstruction mode collapse.** Each row compares training **without** (top row) and **with** (bottom row) deferred quantization on a synthetic mixture-of-Gaussians dataset. **(a) Input dim = 2, Token dim = 1.** Left: data-space scatter plots of original inputs (light blue) and reconstructions (dark blue). Right: kernel density estimates (KDE) of encoder embeddings (light pink) and quantized tokens (dark pink) in the latent space. Without deferred quantization, tokens concentrate in a narrow region, and reconstructions collapse onto a subset of input modes. With deferred quantization, tokens spread across the embedding support, and all modes are faithfully reconstructed. **(b) Input dim = 48, Token dim = 32.** t-SNE projections of data space (left) and latent space (right). The same pattern persists at higher dimensionality: deferred quantization yields broader token coverage and improved reconstruction fidelity across all input modes.

As shown in the top row of Fig. 2 (a) and (b), the learned tokens densely cluster within a limited region of the latent space, and the reconstructed data distribution collapses accordingly. Consequently, reconstructions fail to capture the full modality spectrum of the original data, exhibiting a clear form of mode collapse at the representation level.

**Root cause for coverage failure: clustered initialization from an untrained encoder.** A key contributing factor is the common codebook initialization practice: token embeddings are initialized from the outputs of an untrained encoder, whose embeddings are already concentrated in a narrow region of the latent space. As shown in Fig. 4, the output distribution of an untrained encoder is significantly more concentrated than that of a trained encoder.

To examine how this initialization contributes to coverage failure, we compare embedding distributions produced by trained and untrained encoders on the synthetic dataset. As shown in Fig. 3, we observe that the untrained encoder yields embeddings with fewer distinct peaks and a narrower spread, suggesting fewer distinguishable modes. Since an untrained encoder lacks meaningful feature extraction ability, it maps diverse inputs to similar embeddings, leading to clustered token initialization and insufficient support in the embedding space.

**Intervention for controlled study.** Building on these observations, we hypothesize that initializing tokens using an encoder that has already learned distinctions of data—and therefore produces dispersed embeddings—should mitigate coverage failure and improve generative diversity. This motivates a simple intervention: *train the autoencoder without VQ first, then initialize the codebook using embeddings from the*

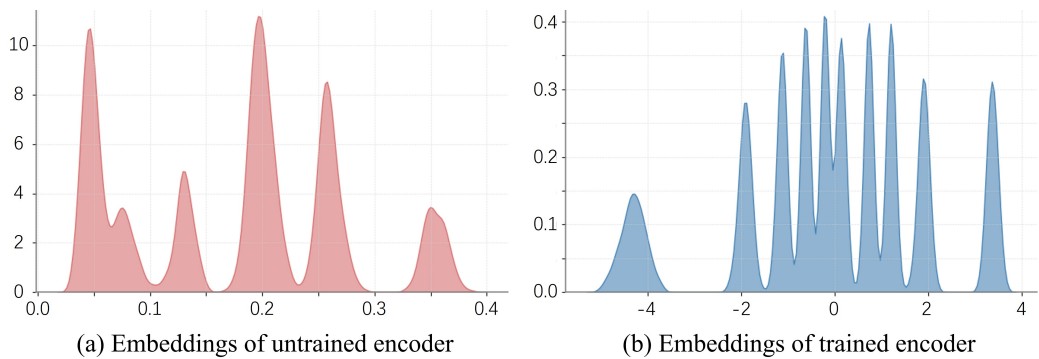

(a) Embeddings of untrained encoder       (b) Embeddings of trained encoder

Figure 3: **Kernel density estimation (KDE) of encoder outputs on the synthetic dataset.** (a) Untrained encoder's output has fewer peaks and a relatively narrow range. (b) Trained encoder's output displays 10 dispersed peaks, which is the same as the input. This phenomenon inspires us that pretraining might be an intervention for token coverage failure.

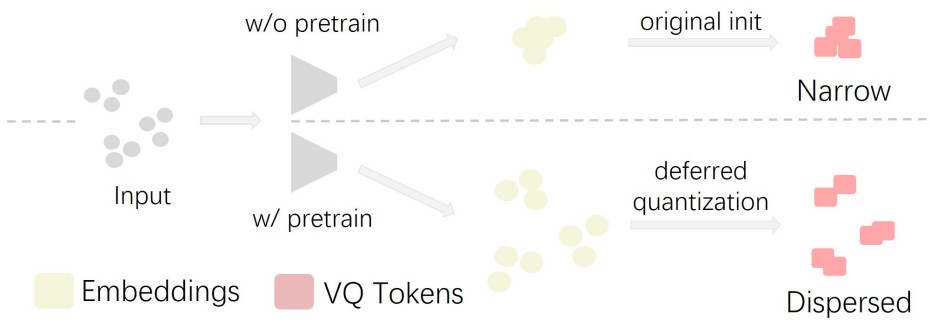

Figure 4: **Early quantization with clustered initialization induces token coverage failure while Deferred Quantization mitigates it.** Initializing the codebook from an untrained encoder yields a narrow, uninformative embedding distribution, causing reduced latent support at early stage. In contrast, Deferred Quantization first learns a dispersed continuous representation and then initializes the codebook with embeddings from the pretrained encoder, yielding better token coverage.

*pretrained encoder.* Concretely, we (i) train an autoencoder, then (ii) train a VQ-VAE initialized with the autoencoder weights.

Pretraining allows the encoder to learn a more structured and spread representation space, providing a more informative foundation for codebook initialization, as shown in Fig. 4.

**Result: mitigating coverage failure improves reconstruction distribution.** We evaluate this intervention on the synthetic dataset (Fig. 2). Comparing subfigures (a) and (b), mitigating token coverage failure yields a more uniform token distribution and a reconstruction distribution that aligns more closely with the original input distribution. Notably, under higher input dimensions (input dim=48), coverage failure leads to a substantially collapsed reconstruction distribution, whereas the mitigated version remains closer to the ground-truth distribution. These results support the view that token coverage failure is a concrete mechanism that can induce mode collapse and reduced diversity.

## 5 Deferred Quantization for Diversity-Preserving Tokenization

This section introduces *Deferred Quantization*, a simple training protocol that decouples early representation learning from discretization to mitigate token coverage failure.

Table 1: **Claim-Evidence map for token coverage failure and Deferred Quantization.** This table summarizes our key theoretical claims and maps them to specific empirical evidence, visualizations, and quantitative metrics provided in the paper.

| ID | Claim | Evidence | Metrics |
|----|-------|----------|---------|
| C1 | Root cause of coverage failure is clustered codebook initialization from untrained encoder biases. | **Fig.2**: Visualization of token coverage failure on synthetic dataset. **Fig.4**: Initialization induced coverage failure. | Latent and reconstruction visualization. |
| C2 | Deferred Quantization reduces coverage failure. | **Sec. 5**: Introduction of Deferred Quantization. **Fig. 2 & Fig. 5 & Tab. 3 & Tab. 5**: Deferred Quantization show better token spread and codebook utilization as well as improved reconstruction quality. | r-FID ↓, LPIPS ↓, MSE ↓, Perplexity ↑, Codebook distance ↑, Vendi Score ↑ |
| C3 | Coverage failure is different from codebook collapse; fixing collapse alone is insufficient. | **Tab. 2**: Deferred Quantization enables gains where collapse-fixing fails. | r-FID ↓, LPIPS ↓, MSE ↓, Perplexity ↑, Codebook distance ↑, Vendi Score ↑ |
| C4 | Generative diversity loss is caused by token coverage failure due to early quantisation, not discretization itself. | **Fig.1**: Illustration of token coverage failure limiting generative diversity. **Tab.4 and Tab. 6**: Quantified loss of diversity (ImageNet-100 & ODIR). | g-FID ↓, Pixel Distance ↑, LPIPS Diversity ↑ |
| C5 | Mitigating poor coverage improves generation quality and diversity across models and datasets. | **Tab. 4 & 6**: Consistent fidelity and diversity gains on ImageNet-100 and ODIR benchmarks. | g-FID ↓, LPIPS Diversity↑, Pixel Distance ↑ |

**Motivation** Token coverage failure is strongly influenced by the state of the encoder embedding space at the time quantization is enabled. When quantization is applied before the encoder learns a sufficiently spread representation, nearest-center assignment can cause poor code initialisation, leading to insufficient coverage.

**Two-stage training protocol** We adopt a two-stage procedure: **Continuous Phase.** Train an autoencoder without vector quantization, allowing the encoder to learn a structured, dispersed embedding space. **Discretization Phase (enable VQ).** Initialize the codebook using pretrained encoder embeddings from the pretrained encoder, then enable vector quantization and continue training under the VQ objective. This protocol encourages the codebook to adapt to an already-formed representation space, thus helping with token coverage.

Deferred Quantization is intentionally minimal: it does not change the generator architecture and introduces no additional inference cost. It directly targets a training-time mechanism (insufficient early coverage) that leads to token coverage failure, thereby improving diversity and reducing mode collapse in token-based generation pipelines.

# 6 Experiments

In this section, we first validate that token coverage failure arises in real-world datasets (CIFAR-10 and ImageNet-100), and then show that coverage failure can propagate to downstream token-based generators as reduced sample diversity and worsened mode coverage (often manifesting as mode collapse). We conduct experiments on two representative token-based generative models, MaskGIT (Chang et al., 2022) and VAR (Tian et al., 2024), using both ImageNet-100 and a medical image dataset. To better elucidate the logical connections between our claims and empirical findings, we provide a claim-evidence map in Tab 1.

It is important to note that in experiments involving generative models, GAN-based losses might introduce smoothing effects that could confound the analysis of token coverage failure. Therefore, in this section, we adopt VQ-VAE as the image tokenizer. The training loss includes $\mathcal{L}_{\text{codebook}}$, $\mathcal{L}_{\text{commitment}}$, $\mathcal{L}_{\text{MSE}}$, and $\mathcal{L}_{\text{LPIPS}}$ (Zhang et al., 2018). Generative experimental results based on VQGAN (Esser et al., 2021) are available in the appendix A.6.

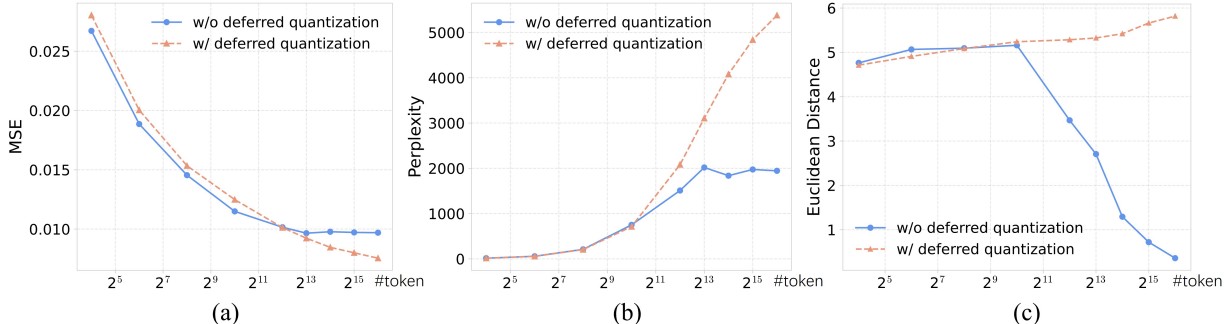

Figure 5: **Deferred Quantization alleviates the token coverage failure on CIFAR-10.** (a) token coverage failure in standard VQ impairs the model's ability to scale, leading to stagnating MSE. Deferred Quantization mitigates this coverage failure, allowing the model to achieve better reconstruction performance as number of token increase. (b)Without Deferred Quantization, perplexity remains low, indicating highly uneven token usage. Deferred Quantization resolves this by spreading representations across the embedding space, ensuring high perplexity and efficient utilization of the available latent capacity. (c) The sharp drop in codebook Euclidean Distance for standard VQ indicates that tokens are clustering into a narrow region. Deferred Quantization maintains a high distance between tokens, preserving codebook coverage.

## 6.1 Setup

**Dataset** As mentioned in Sec. 4, we conduct experiments on a synthetic dataset to validate our hypothesis regarding the causes of token coverage failure. The synthetic dataset consists of 10,000 data points, obtained by sampling 1,000 points from each of 10 Gaussian distributions with identical standard deviations but distinct means. This setup yields ten equally sized classes with similar distribution, designed to emphasize disproportionate token allocation and make token coverage failure patterns more easily observable. To investigate token coverage failure behavior under varying data complexity, we generate synthetic datasets with different input dimensionalities. To further validate the existence of token coverage failure, we additionally conduct experiments on CIFAR-10 and ImageNet-100.

For experiments on generative models, we adopt ImageNet-100, a subset of the ImageNet-1K dataset containing 100 classes. The original ImageNet-100 comprises approximately 130,000 training images and 5,000 test images. To better evaluate both reconstruction-FID (r-FID) and generation-FID (g-FID), we uniformly sample 20,000 images from all training classes to build the test dataset and construct an additional validation set containing 5,000 images.

For the medical domain, we adopt the Ocular Disease Recognition (ODIR) (Maranhão, 2020) dataset, which contains 6,716 fundus images labeled across 8 diagnostic categories. We use a 70%/20%/10% split to partition the data into training, test, and validation sets.

**Metrics** For the synthetic dataset, we directly visualize the original data and its reconstructions, along with the corresponding token and embedding distributions, as shown in Fig. 2. For high-dimensional data, t-SNE is applied for dimensionality reduction prior to visualization.

To quantify token coverage failure, we report the Vendi Score (Friedman & Dieng, 2022) and codebook perplexity as primary indicators of overall codebook coverage, together with average pairwise Euclidean distance and average pairwise cosine distance as auxiliary diagnostics. Following Friedman & Dieng (2022), we compute the Vendi Score using the inner-product kernel on the codebook entries. We emphasize that the Euclidean distance is scale-sensitive and should not be interpreted as a standalone criterion where larger values are always better; instead, it provides a complementary view of the magnitude-based dispersion of codebook vectors, while cosine distance captures their angular dispersion.

To evaluate tokenizer reconstruction performance, we adopt reconstruction FID (r-FID), mean squared error (MSE), and LPIPS scores. For generative quality, we utilize generation FID (g-FID) as the primary metric.

Table 2: **Performance comparison of VAR tokenizers on the ImageNet-100 dataset.** SimVQ(Zhu et al., 2025) is a well-known method to solve the code collapse problem. The results show that mitigating code collapse alone does not lead to a substantial improvement, indicating that token coverage failure is different from code collapse.

| VAR Tokenizer | r-FID ↓ | MSE ↓ | LPIPS ↓ | Euc. ↑ | Cos.↑ | Vendi.↑ | Perp. ↑ |
|---|---|---|---|---|---|---|---|
| w/o Deferred | 5.39 | 2.60 | 1.85 | 1.19 | 0.64 | 2.19 | 2801.88 |
| + SimVQ | 5.52 | 2.50 | 1.78 | 0.80 | 0.96 | 2.07 | 6920.23 |
| w/ Deferred | 5.04 | 2.22 | 1.63 | **7.56** | 0.97 | 2.26 | 7044.51 |
| + SimVQ | **4.93** | **2.17** | **1.59** | 0.93 | **1.00** | **3.47** | **8222.83** |

To assess diversity and distributional coverage of generated samples, we compute the average pairwise pixel-level distance between generated images. To reflect perceptual diversity, LPIPS Diversity is computed as the mean of learned perceptual distances between all pairs of generated images. More details will be available in Appendix A.2.

**Model Architecture** For generative model experiments, we follow the tokenizer framework proposed in VQGAN (Esser et al., 2021). Due to resource limitations, we resize all input images to 128×128 resolution and reduce the backbone's channel size to 64. To preserve a 16×16 latent spatial resolution, one downsampling layer and its upsampling layer are removed. All tokenizer experiments are conducted with a fixed codebook size of 16,384. To ensure feasibility under limited resources, we use the smallest generative model configurations. The MaskGIT generator employs a ViT (Dosovitskiy, 2020) with depth of 24, while the VAR model uses a depth of 16.

**Training Configuration** For the synthetic dataset, the autoencoder is trained for 100 epochs during the Continuous Phase, followed by 100 epochs with quantization enabled during the Discretization Phase. The baseline VQ-VAE is trained for 200 epochs in total to match the training budget. For CIFAR-10, the encoder–decoder is trained for 50 epochs during the Continuous Phase, followed by 250 epochs with quantization enabled during the Discretization Phase. For Deferred Quantization with EMA-based codebook updates Van Den Oord et al. (2017), the model is trained for 150 epochs in the Continuous Phase and 150 epochs in the Discretization Phase. For ImageNet-100 and ODIR, the encoder-decoder is first trained for 200 epochs in the Continuous Phase. After enabling quantization, both our method and the baselines are trained using a ReduceLROnPlateau scheduler with a patience of 20 epochs: the learning rate is halved whenever the validation loss fails to improve within 20 epochs, decaying from $1 \times 10^{-4}$ down to $1 \times 10^{-7}$. Training is terminated via early stopping once no further improvement is observed at the minimum learning rate.

For the LlamaGen tokenizer experiment on ImageNet-1K, we train the tokenizer from scratch and follow the official LlamaGen(Sun et al., 2024) tokenizer training recipe. The tokenizer uses the same architecture as LlamaGen, with a codebook size of 16,384 and latent dimension of 8. For Deferred Quantization, we first train the encoder-decoder without vector quantization for 50,000 steps, corresponding to approximately 5 epochs. We then initialize the codebook from encoder embeddings produced by this continuous model, enable vector quantization, and train for 40 epochs. The baseline LlamaGen number is directly cited from the original paper, while our Deferred Quantization result is obtained under the same architecture and evaluation protocol.

Compared with the 40-epoch LlamaGen baseline, Deferred Quantization introduces an additional continuous autoencoding phase of about 5 epochs. This phase is computationally lighter than VQ training because it does not involve codebook lookup, codebook loss, commitment loss, or EMA codebook updates. While this introduces extra training, our CIFAR-10 ablation in Table 9 suggests that the benefit of the Continuous Phase arises early, supporting the view that the improvement is not merely due to a substantially larger training budget.

## 6.2 Analysis of Coverage Failure in Tokenization

To validate that token coverage failure exists under real-world data conditions, we conduct experiments on CIFAR-10. Additionally, we hypothesize that given a fixed dimensionality of the representation space, increasing the number of tokens tends to facilitate their clustering, thereby making token coverage failure more pronounced and more easily observable. Therefore, we evaluate VQ-VAE performance across varying token quantities.

As shown in Fig. 5, the original VQ model performs well when the number of tokens is relatively small. However, as the token count increases, particularly beyond $2^{12}$, its reconstruction performance deteriorates relative to the pretrained counterpart. Notably, the perplexity curve of the original VQ flattens after $2^{13}$ tokens, indicating poor token utilization. Additionally, its average Euclidean distance remains consistently lower than that of the pretrained model, suggesting a higher degree of similarity among tokens. These findings collectively indicate that coverage failure becomes increasingly severe as the token set grows, leading to reduced coverage of the representation space.

One possible reason why the model utilizing a pretrained AE backbone underperforms the original VQ-VAE at low token numbers is the gap between the continuous representations learned during pretraining and the discrete representations during finetuning, which poses challenges to the VQ learning process. However, this negative impact is outweighed by the benefits of mitigating token coverage failure as the codebook size increases. Overall, pretraining provides an effective intervention for mitigating coverage failure and shows promise in better exploiting large codebooks. It remains an open and valuable direction to explore how to mitigate the performance gap when the number of tokens is small.

We further analyze token coverage failure under the multi-scale tokenizer of VAR on ImageNet-100. To highlight that token coverage failure is distinct from code collapse, we incorporate SimVQ Zhu et al. (2025), an efficient method specifically designed to mitigate code collapse. The results are reported in Tab. 2. We observe that when token coverage failure is present, addressing only code collapse with SimVQ does not yield a substantial improvement in tokenizer performance. In contrast, when both token coverage failure and code collapse are simultaneously resolved, the tokenizer achieves a notable performance gain. This supports the claim that coverage failure is a separate bottleneck beyond dead-token (code-collapse) issue.

## 6.3 Analysis of Coverage Failure in Generation

### 6.3.1 ImageNet-100

**Tokenizer performance.** Both types of original tokenizers exhibit clear token coverage failure, as shown in Tab. 3. For the tokenizer used in MaskGIT (Chang et al., 2022), we observe limited variation among tokens indicated by relatively small Euclidean distances (6.45 *vs.* 18.75), reflecting high similarity between tokens. In addition, the tokenizer exhibits low perplexity (924.57 *vs.* 5311.88), suggesting that only a small subset of tokens is frequently utilized. Together, these observations imply that token usage is poorly aligned with the embedding space, pointing to token coverage failure.

After pretraining, tokens are more evenly utilized and better aligned with the embedding space, indicating that the intervention mitigates token coverage failure. As a result, the pretrained tokenizer achieves improved reconstruction performance, with lower r-FID (8.58 *vs.* 12.22), LPIPS (2.34 *vs.* 2.70), and MSE (3.28 *vs.* 3.91). A similar pattern is observed for the multi-scale tokenizer in VAR: without pretraining, severe coverage is evident, whereas pretraining alleviates it and improves reconstruction.

**Generative performance.** token coverage failure degrades downstream generation, manifesting as both worse sample quality and reduced diversity (often as mode collapse), as shown in Tab. 4. For MaskGIT, poor coverage leads to worse g-FID and reduced pairwise pixel distance and LPIPS diversity among generated samples, indicating that outputs become more similar and cover fewer variations. This aligns with the mechanism that coverage failure cripples the token distribution available to the generator.

For VAR, we observe a consistent pattern: coverage failure in the multi-scale tokenizer leads to reduced generation quality and a drop in diversity. These results reinforce the conclusion that insufficient token

Table 3: **Performance evaluation of various tokenizer on the ImageNet-100 dataset.** ✓ indicates that Deferred Quantization is applied, whereas ✗ indicates that Deferred Quantization is not applied.

| Tokenizer | w/ Deferred? | r-FID ↓ | MSE ↓ | LPIPS ↓ | Euc. ↑ | Cos.↑ | Vendi.↑ | Perp. ↑ |
|---|---|---|---|---|---|---|---|---|
| MaskGIT | ✓ | **8.58** | **3.28** | **2.34** | **18.75** | **0.94** | **2.15** | **5311.88** |
| | ✗ | 12.22 | 3.91 | 2.70 | 6.45 | 0.67 | 1.57 | 924.57 |
| VAR | ✓ | **5.04** | **2.22** | **1.19** | **7.56** | **0.97** | **2.26** | **7044.51** |
| | ✗ | 5.39 | 2.60 | 1.85 | 1.19 | 0.64 | 2.19 | 2801.88 |

Table 4: **Comparison of generative models in the presence and absence of token coverage failure on ImageNet-100.** ✓ indicates that token coverage failure is mitigated, whereas ✗ indicates without mitigation. token coverage failure impairs both generative quality (g-FID) and diversity (pixel distance and LPIPS diversity).

| Model | w/ Deferred? | g-FID ↓ | Pixel Dist. ↑ | LPIPS D. ↑ |
|---|---|---|---|---|
| MaskGIT | ✓ | **14.60** | **80.77** | **0.677** |
| | ✗ | 14.75 | 75.89 | 0.668 |
| VAR | ✓ | **10.44** | **75.75** | **0.667** |
| | ✗ | 12.88 | 70.69 | 0.633 |

coverage limits the model's ability to represent the full data distribution, which propagates to mode collapse / reduced diversity in downstream generation. Qualitative examples are shown in A.4.

### 6.3.2 ODIR Medical Dataset

Table 5: **Performance evaluation of VAR tokenizer on the ODIR medical dataset.** ✓ indicates that token coverage failure is mitigated, whereas ✗ indicates without mitigation. The results indicate that token coverage failure also exists in a real-world medical setting.

| Model | w/ Deferred? | r-FID ↓ | MSE ↓ | LPIPS ↓ | Euc. ↑ | Cos.↑ | Vendi.↑ | Perp. ↑ |
|---|---|---|---|---|---|---|---|---|
| VAR | ✓ | 11.04 | **2.05** | **6.79** | **8.50** | **0.90** | **2.05** | **5396.17** |
| | ✗ | **10.91** | 2.57 | 8.79 | 1.27 | 0.62 | 1.88 | 940.55 |

To further validate our findings, we conduct experiments on the ODIR medical image dataset. For VAR, we again confirm the presence of token coverage failure, as shown in Tab. 5. We also observe a corresponding decline in generative performance, including reductions in both sample quality and diversity (Tab. 6), consistent with our observations on natural image datasets.

We observe that r-FID slightly increases for the VAR tokenizer on ODIR when the coverage failure phenomenon is absent. We attribute this to the relatively small size of the ODIR test set (about 1,200 images), which can introduce variability in FID-based evaluations. As shown in Tab. 5, other quality metrics such as MSE and LPIPS consistently improve after applying our pretraining intervention, indicating clear gains in reconstruction quality. Another contributing factor is that FID relies on features extracted by an Inception-V3 network pretrained on ImageNet-1K, which may not capture semantic differences reliably for domain-specific medical images such as fundus photographs.

However, results for MaskGIT on ODIR do not fully align with expectations. Although the tokenizer shows clear evidence of token coverage failure, generated images do not exhibit a noticeable loss of diversity. One possible explanation is the relatively small size of the ODIR dataset (approximately 6,000 images) together

Table 6: **Generative results of VAR on the ODIR medical dataset.** ✓ indicates that Deferred Quantization is applied, whereas ✗ indicates that Deferred Quantization is not applied. The results indicate that token coverage failure also exists in a real-world medical setting.

| Model | w/ Deferred? | g-FID ↓ | Pixel Dist. ↑ | LPIPS D. ↑ |
|-------|:------------:|:-------:|:-------------:|:----------:|
| VAR | ✓ | **34.33** | **49.83** | **0.401** |
|     | ✗ | 37.65 | 49.01 | 0.390 |

Table 7: **Comparison with established tokenizer baselines under commonly reported** $256 \times 256$ **ImageNet-1K validation settings.** Results marked with $*$ are directly cited from the original paper or official GitHub repository. For codebook coverage metrics of official baselines, we compute the metrics using publicly available checkpoints. Our Deferred Quantization result is obtained by training the LlamaGen tokenizer from scratch under the same architecture and evaluation setting.

| Method | r-FID ↓ | PSNR ↑ | SSIM ↑ | Vendi ↑ | Euc. ↑ | Cos. ↑ |
|--------|:-------:|:------:|:------:|:-------:|:------:|:------:|
| VAR (Tian et al., 2024) | 0.90* | 22.11 | 0.730 | 5.61 | 1.11 | 1.0000 |
| VQGAN (Esser et al., 2021) | 4.98* | 20.00 | 0.629 | 1.50 | 1.59 | 1.0000 |
| LlamaGen (Sun et al., 2024) | 2.19* | 20.79 | 0.675 | 5.29 | 1.42 | 0.9927 |
| + Deferred | 2.12 | 20.82 | 0.676 | 5.38 | 6.48 | 0.9956 |

with its limited variability, as all classes consist of closely related ocular images. Such characteristics may reduce the sensitivity of downstream generative evaluation and mask the impact of coverage failure. Detailed quantitative analyses are included in the supplementary material.

### 6.4 Comparison with Established Tokenizers

To better position our method relative to established tokenizer baselines, we additionally compare with official VQGAN (Esser et al., 2021), LlamaGen (Sun et al., 2024), and VAR (Tian et al., 2024) tokenizer results under commonly reported 256×256 ImageNet-1K validation settings. The results are reported in Table 7. We include the official VQGAN and VAR numbers to provide context against established tokenizers. Since exact reproduction of these official training settings is difficult under our current resources and available code, we use the LlamaGen repository as the controlled and reproducible comparison setting. LlamaGen follows the VQGAN tokenizer design while providing a modern and usable training pipeline.

As shown in Table 7, under this matched LlamaGen setting, Deferred Quantization improves r-FID from 2.19 to 2.12, PSNR from 20.79 to 20.82, and SSIM from 0.675 to 0.676. More importantly, it also improves codebook coverage metrics, increasing Vendi Score from 5.29 to 5.38 and Euclidean distance from 1.42 to 6.48. These results support our main claim that Deferred Quantization mitigates token coverage failure.

We note that the official VAR tokenizer achieves the strongest r-FID, partly because it is trained on Open-Images, which is substantially larger than ImageNet-1K. Moreover, the official VAR tokenizer training code and detailed training recipe are not publicly released, making exact reproduction difficult. Therefore, we include VAR tokenizer mainly to provide context relative to a strong established tokenizer.

To avoid confusion across settings, Table 7 is intended as baseline context rather than a direct extension of Table 3. Table 3 reports controlled ImageNet-100 experiments at 128×128 with a reduced-capacity tokenizer, whereas Table 7 reports established tokenizer comparisons on ImageNet-1K at 256×256. Thus, Table 3 isolates the effect of Deferred Quantization under matched settings, while Table 7 contextualizes it against established baselines.

## 6.5 Ablation Study

Table 8: Ablation on the benefit of the Discretization Phase on CIFAR-10 (codebook size = 8,192). Post-hoc k-means trains the autoencoder for 300 epochs without quantization and applies k-means discretization afterwards. All methods use the same total training budget of 300 epochs.

| Method | MSE ↓ ($\times 10^{-3}$) | Perp. ↑ | Vendi ↑ |
|---|---|---|---|
| Baseline VQ-VAE (300 ep) | 9.6 | 2018.63 | 1.66 |
| Post-hoc k-means (300 ep AE) | 21.0 | 2724.57 | 2.04 |
| Deferred Quant. (150 + 150 ep) | **9.2** | **3104.97** | **1.95** |

**Benefit of the Discretization Phase.** To establish the necessity of jointly training the quantizer with the encoder-decoder, we compare Deferred Quantization against post-hoc k-means discretization on CIFAR-10 with a codebook size of 8,192. We consider three configurations: (1) a standard VQ-VAE trained for 300 epochs, (2) Deferred Quantization with 150 epochs of the Continuous Phase followed by 150 epochs of the Discretization Phase, and (3) an autoencoder trained for 300 epochs without quantization, followed by k-means discretization on the encoder outputs without joint fine-tuning. All methods use the same total training budget of 300 epochs.

As shown in Tab. 8, post-hoc k-means achieves higher codebook diversity (Perplexity and Vendi Score) than the baseline VQ-VAE, confirming that pretraining the encoder before discretization is beneficial for codebook coverage. However, its reconstruction quality degrades substantially (MSE of 21.0 vs. 9.6), because the decoder was trained on continuous representations and cannot directly handle discretized inputs without joint adaptation. Deferred Quantization achieves the best of both worlds: it preserves the diversity gains of pretraining while maintaining reconstruction quality comparable to the baseline, demonstrating that the Discretization Phase—where the quantizer is jointly fine-tuned with the encoder-decoder—is essential for closing the gap between continuous and discrete representations.

**Effect of Continuous Phase Duration.** We further investigate how the duration of the Continuous Phase affects downstream VQ-VAE performance. We fix the Discretization Phase to 150 epochs and vary the Continuous Phase from 0 to 100 epochs on CIFAR-10 with a codebook size of 8,192. Tab. 9 reports the corresponding VQ-VAE performance after the Discretization Phase.

Table 9: VQ-VAE performance initialized by autoencoders with different Continuous Phase durations on CIFAR-10 (codebook size = 8,192). All configurations use 150 epochs for the Discretization Phase. Epoch 0 corresponds to the standard VQ-VAE baseline without pretraining.

| AE Epoch | 0 | 10 | 20 | 40 | 60 | 80 | 100 |
|---|---|---|---|---|---|---|---|
| Perplexity | 2018.63 | 3127.93 | **3143.12** | 3126.99 | 3089.81 | 3107.58 | 3039.83 |
| MSE ($\times 10^{-3}$) | 9.65 | 9.32 | 9.23 | 9.30 | 9.23 | 9.20 | **9.18** |

Firstly, we observed that even a short Continuous Phase (e.g., 10 epochs) is sufficient to improve VQ-VAE performance: MSE drops from 9.65 to 9.32, and perplexity increases from 2018.63 to 3127.93, indicating that the encoder quickly learns a dispersed representation that benefits codebook initialization. Second, beyond approximately 20 epochs, continued autoencoder training yields only marginal gains in downstream VQ-VAE performance, despite the autoencoder's own validation loss continuing to decrease. This suggests that the benefits of the Continuous Phase plateau once the encoder has established a sufficiently well-distributed representation space, and a moderate pretraining duration is already sufficient to mitigate token representation shrinkage.

Although this ablation is conducted on CIFAR-10 rather than ImageNet-1K, it suggests that the benefit of the Continuous Phase arises early and is not solely due to a substantially larger training budget.

Table 10: Comparison of deferred quantization with variance-scaled codebook initialization. All baselines apply quantization from the start of training. Best results are in **bold**.

| Method | MSE ↓ | Perplexity ↑ | Vendi. ↑ |
|---|---|---|---|
| Gaussian Init ($\sigma = 1.0$) | 9.6 | 1889.0 | 1.579 |
| Gaussian Init ($\sigma = 2.0$) | 9.8 | 1777.5 | 1.552 |
| Gaussian Init ($\sigma = 3.0$) | 9.8 | 1930.0 | 1.557 |
| Gaussian Init ($\sigma = 10.0$) | 9.6 | 2207.4 | 1.577 |
| **Deferred Quantization (Ours)** | **8.5** | **4079.4** | **1.953** |

**Comparison with Variance-scaled Initialization.** A natural question is whether the benefits of deferred quantization can be replicated by simply initializing the codebook with larger variance, thereby encouraging broader initial coverage of the latent space without a separate pretraining stage. To investigate this, we compare our method against baselines in which codebook vectors are drawn from $\mathcal{N}(0, \sigma^2 I)$ with $\sigma \in \{1.0, 2.0, 3.0, 10.0\}$, while quantization is applied from the start of training. Results are reported in Table 10.

Across all initialization scales, codebook diversity remains largely unchanged: the Vendi Score stays around 1.55–1.58, even at $\sigma = 10.0$ where codebook vectors are widely dispersed initially. The MSE similarly shows only marginal variation (0.0096–0.0098). We hypothesize that this issue is not primarily caused by insufficient initial spread of the codebook vectors, but rather by their misalignment with the encoder's output. Since randomly dispersed vectors are unrelated to the encoder's output distribution, nearest-neighbor assignment tends to pull them back toward the narrow early encoder distribution during training, effectively erasing the initial spread. In contrast, Deferred Quantization reduces the MSE to 0.0085 and raises the Vendi Score to 1.953, indicating that token coverage failure is mitigated and leading to improved codebook diversity.

## 6.6 Comparison with VQ Optimization Methods

Table 11: Comparison with VQ optimization methods on CIFAR-10 across codebook sizes. Methods from VQSTE++ (Huh et al., 2023) and SoundStream (Zeghidour et al., 2021) are implemented based on official repository `vqtorch`. *Affine + Sync (default) training failure at codebook size 1,024.

| Method | MSE ↓ ($\times 10^{-3}$) | | | | | Perplexity ↑ | | | | | Vendi. ↑ | | | | |
|---|---|---|---|---|---|---|---|---|---|---|---|---|---|---|---|
| | 1K | 4K | 8K | 16K | 32K | 1K | 4K | 8K | 16K | 32K | 1K | 4K | 8K | 16K | 32K |
| Baseline | 12.39 | 10.99 | 10.15 | 10.00 | 9.80 | 753.67 | 1481.82 | 2489.56 | 2235.81 | 2532.43 | 1.89 | 1.71 | 1.66 | 1.55 | 1.50 |
| Baseline + EMA | **11.49** | 10.15 | 9.65 | 9.77 | 9.72 | 750.94 | 1509.93 | 2018.63 | 1837.66 | 1974.73 | **2.03** | 1.80 | 1.70 | 1.58 | 1.52 |
| Affine + Sync* | – | 13.26 | 11.65 | 12.15 | 11.93 | – | 460.33 | 859.37 | 738.97 | 819.92 | – | 1.53 | 1.58 | 1.50 | 1.47 |
| Affine (affine_lr=5) | 12.06 | **9.86** | 10.23 | 10.17 | 9.43 | 783.65 | 2239.34 | 2011.03 | 1941.82 | 2999.24 | 1.86 | 1.82 | 1.63 | 1.53 | 1.53 |
| Sync (sync_nu=0.1) | 12.17 | 11.53 | 11.43 | 11.10 | 10.98 | 726.15 | 1098.52 | 988.92 | 1110.11 | 1306.09 | 1.92 | 1.65 | 1.53 | 1.50 | 1.47 |
| LRU + K-means | 12.24 | 10.00 | 9.20 | 8.62 | 8.32 | **786.35** | **2341.85** | **3777.72** | **5188.40** | **6030.98** | 1.87 | 1.83 | 1.81 | 1.77 | 1.75 |
| Deferred (Ours) | 12.32 | 10.26 | 9.35 | 8.75 | 8.15 | 637.82 | 1786.02 | 2712.89 | 3671.48 | 4548.58 | 1.89 | 1.96 | 1.93 | **1.97** | 1.90 |
| Deferred + EMA (Ours) | 12.49 | 10.13 | **9.23** | **8.46** | **8.00** | 712.86 | 2082.09 | 3104.97 | 4079.40 | 4839.72 | 1.98 | **1.97** | **1.95** | 1.95 | **1.99** |

To address concerns regarding comparisons with prior VQ optimization approaches, we conduct experiments on CIFAR-10 across codebook sizes ranging from 1,024 to 32,768.

**Baselines.** We compare Deferred Quantization against the following methods: (1) **VQ-VAE** (Van Den Oord et al., 2017): the standard VQ-VAE with commitment loss; (2) **EMA** (Van Den Oord et al., 2017): VQ-VAE with exponential moving average codebook updates; (3) **Affine, Sync** (Huh et al., 2023): affine parameterization and synchronous update from VQ-STE++. We implement all three techniques proposed in Huh et al. (2023) using the official `vqtorch` library. However, Alternated Optimization consistently led to training collapse across different settings. We hypothesize that this may be due to differences in library dependencies and hardware configurations that have evolved since the original publication. For the Affine

+ Sync combination, we use the default parameters provided in `vqtorch` (affine_lr = 2, sync_nu = 2). We additionally conduct hyperparameter tuning for Affine Parameterization and Synchronous Update, respectively, yielding affine_lr = 5 and sync_nu = 0.1, and report results for both configurations; (4) **LRU + K-means**: the codebook maintenance strategy from SoundStream (Zeghidour et al., 2021), which initializes the codebook from the first batch of training data and replaces dead tokens based on a least-recently-used (LRU) policy during training.

**Results.**  As shown in Tab. 11, we show the findings below:

*Deferred Quantization scales favorably with codebook size.* At small codebook sizes (1,024), Deferred Quantization performs comparably or slightly inferior to other methods. However, as the codebook size increases, its reconstruction quality improves substantially, becoming comparable to LRU + K-means (Zeghidour et al., 2021) at 8,192 codes and surpassing it at larger sizes. This is consistent with our analysis in Sec. 6.2: early quantization causes token representation shrinkage that becomes increasingly severe as the codebook grows, and Deferred Quantization directly mitigates this effect.

*Deferred Quantization + EMA achieves the best overall reconstruction.* At codebook sizes greater than 8,192, the combination of Deferred Quantization with EMA updates consistently achieves the lowest MSE across all methods, outperforming both the baselines and the methods from Huh et al. (2023) and Zeghidour et al. (2021).

*Deferred Quantization maintains consistently high Vendi Scores.* Regardless of codebook size, both Deferred Quantization and Deferred Quantization + EMA maintain Vendi Scores around 1.90–1.99, higher than all other methods. This indicates that our approach preserves greater latent space diversity across codebook sizes. In contrast, other methods (including LRU + K-means) show declining Vendi Scores as codebook size increases, reflecting reduced diversity of the latent space despite improved reconstruction. This supports our central claim that Deferred Quantization mitigates token coverage failure, not merely at the codebook utilization level.

## 7   Conclusion & Limitation

In this work, we systematically analyze *token coverage failure* in vector quantization as a critical and previously underappreciated factor that can drive reduced diversity and mode collapse in token-based generative models. We show that a commonly adopted codebook initialization strategy—initializing tokens from an untrained encoder—induces poor coverage in the token embedding space, leading to unbalanced token usage and insufficient latent support for downstream generation.

A limitation of our current evaluation is that we cannot exactly reproduce the official VAR tokenizer training setting, because its tokenizer training code and detailed recipe are not publicly released and the full-scale training setting requires substantially more computational resources than available to us. Therefore, we do not claim to outperform the official VAR tokenizer under its full-scale OpenImages training setting. Instead, our claim is that Deferred Quantization improves reconstruction and codebook coverage under matched and reproducible settings, as demonstrated by the LlamaGen comparison.

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

# A  Appendix

## A.1  Implementation details

**Sythentic Dataset**  Our synthetic dataset includes 10 clusters, each with 1,000 data points sampled from a Gaussian distribution and standardized using Standard Scaler. Our VQ-VAE comprises an MLP-based encoder/decoder with three linear layers and uses the ReLU activation function. Training is facilitated by the AdamW optimizer, with a learning rate of 0.001. Additional specifications include a codebook size of 128, a hidden dimension of 32, a batch size of 256, a beta of 0.25, and a decay rate ($\gamma$ for EMA) of 0.9. In experiments, the autoencoder is trained for 100 epochs. The fine-tuned VQ-VAE and the original VQ-VAE are trained for 100 and 200 epochs respectively.

**CIFAR-10**  For CIFAR-10, our VQ-VAE adopts downsampling using a CNN with a downsample channel of 128, and the model includes two residual blocks with a hidden channel size of 64. The codebook size is set at 512 with a token dimension of 32. The learning rate is 3e-4, using the Adam optimizer with amsgrad set to true. The beta is 0.25 and the decay rate is 0.99. The codebook size in experiments is varied from 16 to 65,536, with embedding and token adopting the size of 32. And we pretrain AE for 150 epochs and fine-tune the VQ-VAE for 150 epochs.

**ImageNet-100 & ODIR**  As described in the main text, we modified the tokenizer by removing one downsampling layer along with its corresponding upsampling layer and reducing the backbone's channel size to 64. We employed both ReduceLROnPlateau and Cosine annealing schedulers to train the tokenizer. Detailed configurations can be found in the provided codebase (*config.yaml*). The initial learning rate was set to $1 \times 10^{-4}$, with a minimum of $1 \times 10^{-6}$, and early stopping was applied. For ImageNet-100, we trained selected the checkpoint with the best FID on the validation set for downstream tasks on ImageNet-100, while the final checkpoint was used for the ODIR dataset. VAR and MaskGIT were trained with ReduceLROnPlateau scheduling and early stopping.

It is important to note that when the codebook size is large, it is infeasible to initialize it using embeddings from a single batch. Therefore, on CIFAR-10, ImageNet-100, and ODIR, we initialize the codebook using embeddings collected from multiple batches. Specifically, we maintain an embedding-to-token ratio of 10:1 on CIFAR-10, and 2:1 on ImageNet-100 and ODIR. Moreover, to improve computational efficiency, we compute LPIPS diversity using a 5,000-image subset of the ImageNet-100 test set. For the ODIR dataset, we use the full original test set.

All tokenizers and generative models are trained on 2 A100 GPUs with 40 GB memory. Training the tokenizers on ImageNet-100 typically takes 1.5 to 3 days, while training the generative models requires 3–6 days depending on the setting.

**ImagneNet-1K**  For the LlamaGen tokenizer results in Table 7, we train the tokenizer from scratch following the official LlamaGen tokenizer training recipe. We use the same tokenizer architecture, codebook size of 16,384, and latent dimension of 8. For Deferred Quantization, we first train the encoder-decoder without vector quantization for 50,000 steps( 5epochs) as the Continuous Phase. We then initialize the codebook using first training batch of embeddings from the pretrained encoder, enabling vector quantization for 40 epochs in the Discretization Phase until convergence. The baseline LlamaGen results are directly cited from the original LlamaGen paper, while our Deferred Quantization result is obtained under the same architecture and evaluation setting. For generative results on ImageNet-1K, we apply the LlamaGen-B model and train it for 300 epochs. The results are available in Appendix A. A.5.

**Initialization Strategy**  For codebook initialization, a widely used initialization strategy is K-means(Zeghidour et al., 2021). It uses the encoder output $\mathcal{Z}$ and perform K-means algorithm to initialize the tokens $\mathcal{T}$, where $N$ is the number of encoder output and $S$ is the number of tokens. The initialization aims to minimize the total distance from each vector $z_j$ to its nearest token $t_k$. The optimizing function is shown in equation 3,

$$\min \sum_{j=1}^{N} \sum_{k=1}^{S} r_{jk} \|z_j - t_k\|^2, \tag{3}$$

where $r_{jk} = 1$ if $z_j$ is assigned to cluster center $t_k$, otherwise $r_{jk} = 0$ .

To quantify token coverage failure, we report three complementary codebook diversity metrics along with perplexity.

## A.2   Metrics for latent space

**Euclidean distance.**   We compute the mean pairwise $L_2$ distance across all codes in the codebook:

$$D_{\text{Euc}} = \frac{2}{S(S-1)} \sum_{i<j} \|t_i - t_j\|_2, \tag{4}$$

where $S$ is the codebook size and $t_i$ denotes the $i$-th code vector. This serves as an indicator of token clustering, with lower values suggesting that code vectors have concentrated in a limited region.

**Cosine distance.**   Since $D_{\text{Euc}}$ is sensitive to the overall scale of the embedding space, we additionally report the mean pairwise cosine distance:

$$D_{\text{cos}} = \frac{2}{S(S-1)} \sum_{i<j} \left( 1 - \frac{t_i^{\top} t_j}{\|t_i\| \, \|t_j\|} \right). \tag{5}$$

**Vendi Score.**   To capture overall codebook diversity, we adopt the Vendi Score (Friedman & Dieng, 2022). Following their formulation, we construct the similarity matrix via the inner-product kernel $K_{ij} = t_i^{\top} t_j$ and compute

$$\text{VS}(T) = \exp\left( -\sum_i \lambda_i \log \lambda_i \right), \tag{6}$$

where $\lambda_i$ are the eigenvalues of $K/\text{tr}(K)$. The Vendi Score is scale-invariant and admits a direct interpretation as the *effective number of distinct codes* in the codebook, taking values in $[1, S]$.

**Perplexity.**   Finally, we report codebook perplexity, computed as the exponential of the entropy over code usage likelihood. It reflects the effective number of tokens being utilized and is maximized when all tokens are used uniformly.

## A.3   Loss curves

Fig. 6 reports the validation loss of different methods on CIFAR-10 with a codebook size of 16384, while Figs. 7 and 8 show the training curves of the failure cases.

Validation loss over epochs

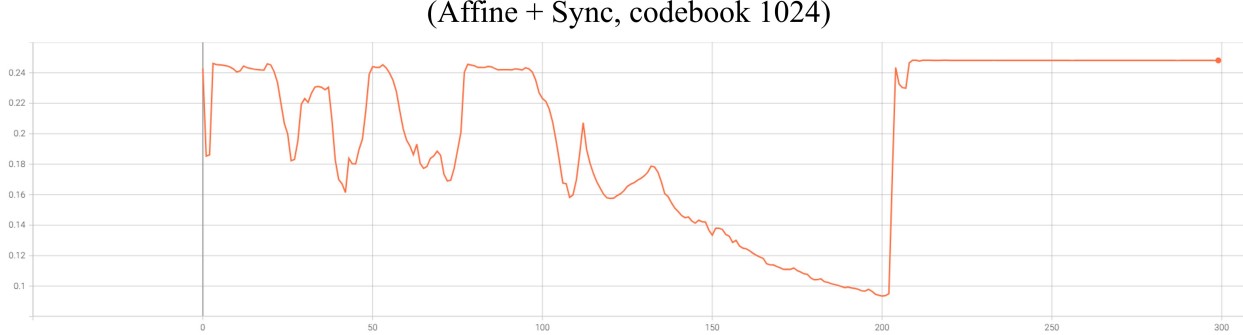

Figure 6: The validation loss curves on CIFAR-10 with the codebook sizes of 16384.

Training loss curve of failure case
(Affine + Sync, codebook 1024)

Figure 7: Training loss curve of failure case(Affine, Sync, codebook 1024).

## A.4    Visualization of generated examples

As mentioned in previous section, for VAR generative model, we observe a consistent pattern: poor coverage in its multi-scale tokenizer leads to reduced generation quality and a drop in diversity. These results reinforce the conclusion that insufficient token coverage limits the model's ability to represent the full data distribution, which propagates to mode collapse / reduced diversity in downstream generation. Qualitative examples are shown in Fig. 9.

## A.5    LlamaGen generative results

Table 12: **Generative results with LlamaGen-B on ImageNet-1K at** $256{\times}256$ **resolution.** We follow the evaluation protocol and metrics used in the LlamaGen (Sun et al., 2024). The generator architecture and training recipe are kept unchanged; the only change is replacing the original tokenizer with our Deferred tokenizer.

| Method | g-FID ↓ | IS ↑ | Precision ↑ | Recall ↑ |
|---|---|---|---|---|
| LlamaGen-B | 5.46 | 193.61 | **0.83** | 0.45 |
| + Deferred tokenizer | **5.07** | **194.81** | **0.83** | **0.47** |

We further provide downstream generative results in Table X. For a fair and comprehensive comparison, we follow the evaluation protocol and metrics used in the LlamaGen paper, including g-FID, Inception Score (IS), and Precision & Recall(Sajjadi et al., 2018).

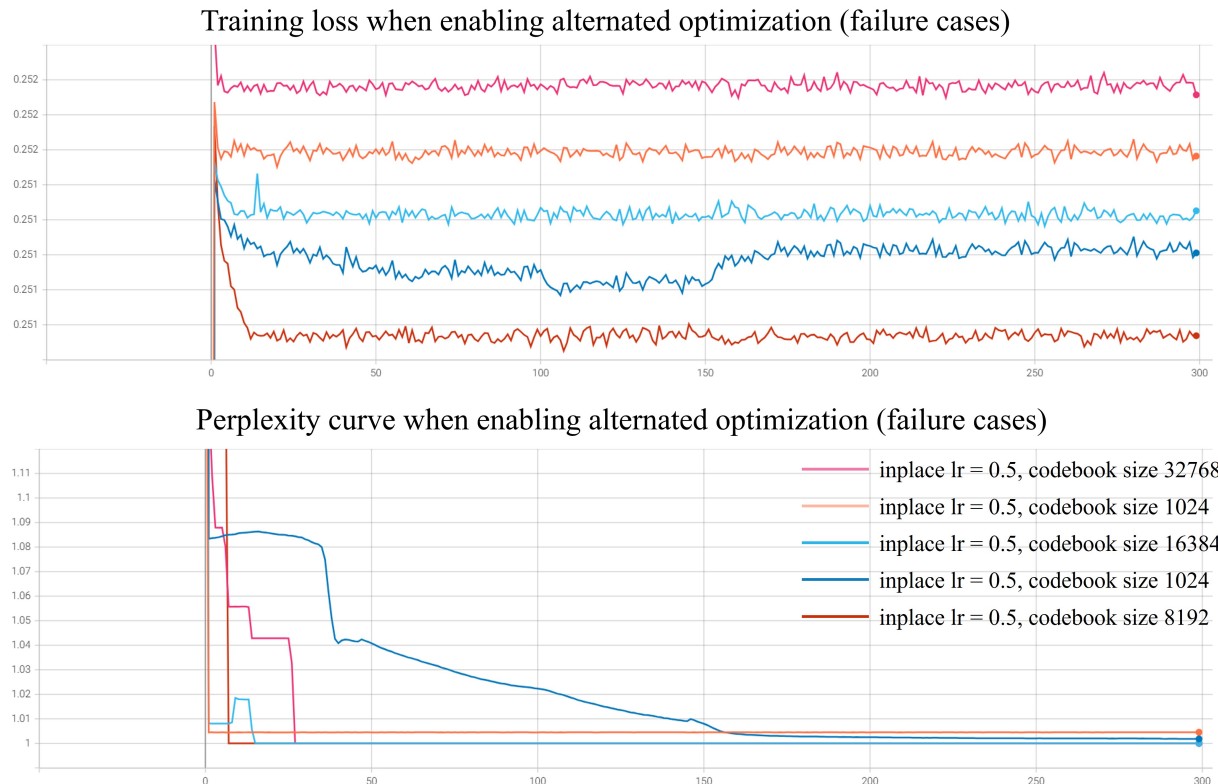

Figure 8: **Failure cases of alternated optimization.** We find that enabling alternated optimization(Huh et al., 2023)leads to codebook collapse across different learning rates and codebook sizes — the training loss stagnates while the perplexity quickly drops toward 1, indicating that only a small fraction of codes remain active. We hypothesize that this instability stems from changes in library dependencies and hardware environments since the method was originally published.

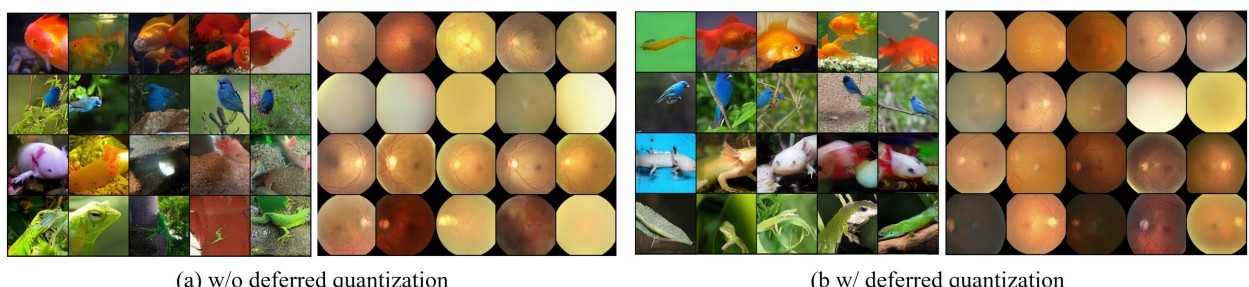

(a) w/o deferred quantization                (b w/ deferred quantization)

Figure 9: **Images generated using VAR.** (a) ImageNet (a.left) and real-world medical images of eyes (a.right) generated by VAR **w/o deferred quantization**. (b) ImageNet (b.left) and real-world medical images of eyes (b.right) generated by VAR **w/ deferred quantization**.

The results show that the improvements at the tokenizer level also propagate to downstream image generation. Compared with the original LlamaGen-B tokenizer, using our Deferred tokenizer improves g-FID from 5.46 to 5.07 and Recall from 0.45 to 0.47, while maintaining the same Precision of 0.83 and slightly improving IS from 193.61 to 194.81. The improvement in Recall suggests that the generator covers a broader portion of the real data distribution, which is consistent with our claim that better token coverage helps improve generative diversity.

### A.6 VQGAN-based Generation Results

Table 13: **Comparison of generative models based on VQGAN on ImageNet-100.** ✓ indicates that token coverage failure is mitigated, whereas ✗ indicates without mitigation.

| Model | w/ Deferred? | g-FID ↓ | Pixel Dist. ↑ | LPIPS D.↑ |
|---|---|---|---|---|
| MaskGIT | ✓ | **10.85** | **79.17** | 0.712 |
| | ✗ | 12.25 | 78.83 | **0.714** |
| VAR | ✓ | 8.30 | **77.10** | **0.719** |
| | ✗ | **7.83** | 73.37 | 0.714 |

Table 14: **ODIR generation based on VQGAN of VAR.** ✓ indicates that token coverage failure is mitigated, whereas ✗ indicates without mitigation. Token coverage failure impairs generative quality and diversity.

| Model | w/ Deferred? | g-FID ↓ | Pixel Dist. ↑ | LPIPS D.↑ |
|---|---|---|---|---|
| VAR | ✓ | 29.65 | **50.56** | 0.550 |
| | ✗ | **27.89** | 45.14 | **0.570** |

As shown in Tab.15 and Tab.16, low Euclidean distance and low perplexity indicate the presence of token coverage failure in VQGAN. And token coverage failure still degrades reconstruction quality. For generative models, Tab.13 and Tab.14 demonstrate the impact of token coverage failure might affect generative diversity and fidelity. In particular, Pixel Distance is consistently lower under poor coverage, reflecting reduced pixel diversity in generated samples. However, we found better r-FID and g-FID scores in the presence of poor coverage. We hypothesize that this is due to GAN loss might enhance the decoder's expressive capacity, which enables it to compensate for the effects of token coverage failure.

### A.7 Results for MaskGIT on Medical dataset

As shown in Tab. 17, when token coverage failure occurs, the tokenizer in MaskGIT also exhibits a decline in reconstruction quality on the ODIR dataset. However, the generative model unexpectedly displays high quality and diversity under token coverage failure. We hypothesize that this counterintuitive result may stem from the limited data available for training and evaluation. The ODIR dataset contains only about 6,000 images, all of which are restricted to ocular imagery, resulting in lower diversity.

Table 15: **Evaluation of tokenizers trained with GAN loss on the ImageNet-100 dataset.** ✓ indicates that token coverage failure is mitigated, whereas ✗ indicates without mitigation. MSE and LPIPS values in the table are scaled. To recover the actual values, multiply them by $10^{-4}$ and $10^{-3}$, respectively.

| Tokenizer | w/ Deferred? | r-FID ↓ | MSE ↓ | LPIPS ↓ | Euc. ↑ | Cos. ↑ | Vendi. ↑ | Perp. ↑ |
|---|---|---|---|---|---|---|---|---|
| MaskGIT | ✓ | **5.28** | **3.96** | **2.40** | **18.75** | **0.96** | **2.11** | **5575.92** |
|  | ✗ | 6.40 | 4.58 | 2.70 | 6.52 | 0.67 | 1.57 | 920.95 |
| VAR | ✓ | 2.13 | **2.59** | **1.73** | **7.50** | **0.97** | **2.26** | **7143.41** |
|  | ✗ | **2.09** | 3.00 | 1.87 | 1.13 | 0.66 | 2.10 | 2814.91 |

Table 16: **Evaluation of VAR tokenizers trained with GAN loss on the ODIR dataset.** ✓ indicates that token coverage failure is mitigated, whereas ✗ indicates without mitigation. To recover the actual MSE and LPIPS, multiply them by $10^{-5}$ and $10^{-4}$, respectively.

| Tokenizer | w/ Deferred? | r-FID ↓ | MSE ↓ | LPIPS ↓ | Euc. ↑ | Cos.↑ | Vendi.↑ | Perp. ↑ |
|---|---|---|---|---|---|---|---|---|
| VAR | ✓ | **6.90** | **2.19** | **7.15** | **8.50** | **0.90** | **2.05** | **5451.76** |
|  | ✗ | 8.12 | 2.66 | 9.39 | 1.27 | 0.62 | 1.88 | 801.07 |

Table 17: **MaskGIT tokenizer and generative performance on medical dataset.** $\mathcal{L}_{\text{GAN}}$ denotes whether GAN loss was used during training. ✓ indicates that token coverage failure is mitigated, whereas ✗ indicates without mitigation. To recover the actual MSE and LPIPS values, multiply them by $10^{-5}$ and $10^{-4}$.

| $\mathcal{L}_{\text{GAN}}$ | w/ Deferred? | r-FID ↓ | MSE ↓ | LPIPS ↓ | Euc. ↑ | Perp. ↑ | g-FID ↓ | Pixel Dist. ↑ | LPIPS D. ↑ |
|---|---|---|---|---|---|---|---|---|---|
| No | ✓ | **10.33** | **3.16** | **0.90** | **12.70** | **3346.04** | 38.04 | 48.44 | 0.370 |
|  | ✗ | 11.96 | 4.19 | 1.16 | 4.92 | 438.36 | **27.95** | **49.73** | **0.397** |
| Yes | ✓ | **10.47** | **3.67** | **1.02** | **12.60** | **3211.72** | 38.30 | 48.15 | 0.371 |
|  | ✗ | 12.72 | 4.64 | 1.30 | 4.93 | 432.07 | **32.60** | **49.28** | **0.402** |

