# OpenReview forum: "Early Quantization Shrinks Codebook: A Simple Fix for Diversity-Preserving Tokenization"
_TMLR — Rejected by TMLR_

### Review · Reviewer_pFXH · 2026-03-20

**Summary Of Contributions:**

The authors propose an improvement to VQVAE training. The fix is essentially "pretrain an authorencoder, then finetune with VQ" — this is folk wisdom in the VQ community and has appeared in practice if not always formalized as a standalone contribution.
Usually, this strategy is thought to benefit code usage by reducing dead tokens through better initialization.


The interesting point the authors make is that this method is more than just improving code usage. Even when code usage appears healthy, the codebook can still occupy a geometrically narrow region of latent space that restricts the representational support available to downstream generators. The authors argue that the primary benefit of deferred quantization is not resurrecting dead tokens but rather expanding the latent support of the codebook, and that this geometric breadth, not token utilization rate, is the proximate cause of generative diversity. This reframing is the paper's most interesting conceptual contribution, and it motivates a new set of diagnostics e.g., reconstruction modality coverage that go beyond perplexity as the standard measure of codebook health.

**Audience:**

Yes

**Audience Explanation:**

Yes, the findings are likely to be of interest to a meaningful segment of the TMLR audience. The interesting point the authors make is that this method is more than just improving code usage but also mitigating representation shrinkage.

**Broader Impact Concerns:**

No significant broader impact concerns apply to this work.

**Claims And Evidence:**

Yes

**Claims Explanation:**

The experiments are generally convincing, especially the authors provide additional details on the training configurations. 2 generative models (MaskGIT and VAR) are validated, which is good.
There’s still room for improvement because it does not compare to established baselines, e.g., official VQ-GAN to validate the codec performance. That’s a pitfall.

**Requested Changes:**

1. The paper fails to compare to established baselines, e.g., official VQ-GAN to validate the codec performance. The same for its generative models. Adding these baselines would significantly strengthen the paper.
2. Lack of review on previous papers that apply the same method. e.g, Łańcucki et al. (2020) Robust Training of Vector Quantized Bottleneck Models explicitly proposes a warm-up phase where "during the first M iterations we perform no quantization, letting the network initially stabilize outputs of the encoder,". This is almost identical to Deferred Quantization in mechanism and motivation. There are many other identical practices for training VQVAEs.

---

> ### Author Response · Authors · 2026-04-18
>
> # Reviewer 3 pFXH
>
> > The paper fails to compare to established baselines, e.g., official VQ-GAN to validate the codec performance. The same for its generative models. Adding these baselines would significantly strengthen the paper.
> >
>
> We thank the reviewer for this suggestion. We come from an academic institution and only have access to 4 GPUs. As a result, we are still running experiments and cannot obtain the full generative model results in time.
>
> To address the reviewer's concern, we validate Deferred Quantization on LlamaGen, an established, publicly available tokenizer with relatively lower training cost compared to VAR and MaskGIT. Currently, we have completed the tokenizer training, and the results are shown in Table below.
>
> Table 1. Reconstruction performance across Imagenet 1K 256 × 256
>
> | Method | r-FID ↓ | PSNR ↑ | SSIM | Euc.  | Cos.  | Vendi.  |
> | --- | --- | --- | --- | --- | --- | --- |
> | LlamaGen | 2.19* | 20.79* | 0.675* | 1.42** | 0.9927** | 5.29** |
> | w/ Deferred | **2.12** | **20.82** | **0.676** | **6.48** | **0.9956** | **5.38** |
>
> * Results directly cited from the original LlamaGen paper. ** Results are computed by public checkpoints
>
>  Deferred Quantization consistently outperforms the baseline across all metrics. The Euclidean distance increases from 1.42 to 6.48, and the Vendi Score improves from 5.29 to 5.38, indicating that the codebook captures a broader and more diverse set of representations after applying Deferred Quantization. These results demonstrate that our findings generalize to an established tokenizer framework beyond the reduced-capacity setting used in our main experiments.
>
> We will update the manuscript with full generative model results once training is completed.
>
> > C2: Lack of review on previous papers that apply the same method. e.g, Łańcucki et al. (2020) Robust Training of Vector Quantized Bottleneck Models explicitly proposes ...
> >
>
> Thank you for pointing this out. We agree that our original Related Work did not sufficiently discuss prior studies that use warm-up or pretraining before quantization. In the revised manuscript, we have added a discussion of these related works in the Related Work section and clarified how our contribution differs in focus.

---

> > ### Comment · Reviewer_pFXH · 2026-04-20
> >
> > In table 2, deferred + simvq has the best performance. It has verylow euclidean distance. This shows that euclidean distance is not "the larger the better ". So why do you make that claim?

---

> > > ### Author Response · Authors · 2026-04-20
> > >
> > > We thank the reviewer for this important observation. We agree that Euclidean distance should not be interpreted as universally “the larger the better,” and we will revise the wording accordingly. Specifically, we will clarify in the manuscript that Euclidean distance is a scale-sensitive auxiliary diagnostic, rather than a standalone criterion for codebook quality.
> > >
> > > In our paper, the Vendi Score is the primary metric for overall codebook diversity / coverage, while Euclidean and cosine distances are auxiliary diagnostics that capture the magnitude-based and angular aspects of codebook dispersion, respectively. These diversity metrics are used to assess codebook coverage and to examine whether token coverage failure is mitigated.

---

> > ### Author Response · Authors · 2026-04-23
> >
> > **Update:** We have now completed the generative model training. As shown in Table 2 below, the LlamaGen-B generator trained on our Deferred Quantization tokenizer consistently outperforms the baseline.
> >
> > **Table 2. Generation performance on ImageNet-1K 256 × 256 (LlamaGen-B)**
> >
> > | Method | g-FID ↓ | IS ↑ | Precision ↑ | Recall ↑ |
> > | --- | --- | --- | --- | --- |
> > | LlamaGen-B | 5.46 | 193.61 | 0.83 | 0.45 |
> > | + Deferred tokenizer | **5.07** | **194.81** | **0.83** | **0.47** |
> >
> > The generative results indicate that the improvements at the tokenizer level (Table 1) propagate to downstream generation quality. The g-FID improves from 5.46 to 5.07, and Recall increases from 0.45 to 0.47, indicating that the generator produces samples covering a broader portion of the real data distribution.
> >
> > These results are obtained using a training configuration that strictly follows the original LlamaGen-B setup without any modifications. The consistent improvements on this established full-scale baseline (ImageNet-1K, 256 × 256) demonstrate that our findings generalize beyond the reduced-capacity settings used in our main experiments.

---

> ### Comment · Reviewer_pFXH · 2026-04-30
>
> I'm also confused by the lack of consistency in the r-FID metrics in the paper. In Table 3, the metrics are around 5. Why does it change to a round 2 in your added experiment table 11? Is it just from the tokenizer architecture itself, or do you use a different dataset? If so, why do you use a different dataset?
>
>
> Also, why the authors choose to add the new experient to table 11 instead of adding them to table 3?

---

> ### Comment · Reviewer_pFXH · 2026-04-30
>
> > We thank the reviewer for this suggestion. We come from an academic institution and only have access to 4 GPUs. As a result, we are still running experiments and cannot obtain the full generative model results in time.
>
> I believe instead of avoiding putting official results as baselines, the authors should at least discuss with a paragraph and a table to show the performance differences with official VAR and VQGAN, and why is it (e.g., dataset size? training epoch difference? Or how much compute difference?). Currently it seems the authors are reluctant to compare with official VAR tokenizer and official VQGAN. For me this is not a good sign for credibility. While I appreciate the authors' work on adding substantially new details, I will lean to reject if the authors still does not comment on established baselines.

---

> ### Author Response · Authors · 2026-04-30
> **Response to r-FID**
>
> We thank the reviewer for pointing this out. The r-FID values in Table 3 and Table 11 are not directly comparable because they are obtained under different experimental settings.
>
> Table 3 reports our main controlled experiments on ImageNet-100 using 128×128 images and a reduced-capacity tokenizer backbone, where the channel size is set to 64 due to our limited computational resources. These experiments are designed to compare the effect of Deferred Quantization under the same controlled setting.
>
> Table 11, in contrast, was added to address the reviewer’s request for comparison with an established tokenizer baseline(under Imagenet-1k, 256×256, VQGAN). Therefore, we follow the standard LlamaGen tokenizer setting and report results against the official LlamaGen baseline. The baseline r-FID in Table 11 is directly cited from the original LlamaGen paper, and our Deferred Quantization result is evaluated under the same LlamaGen tokenizer setting for a fair comparison within that table.
>
> Thus, the change from an r-FID around 5 in Table 3 to around 2 in Table 11 is not caused solely by the tokenizer architecture itself, nor should it be interpreted as a direct improvement across tables. It mainly reflects the change in experimental protocol, including dataset setting, image resolution, tokenizer capacity, and the baseline/evaluation setup.
>
> We placed the LlamaGen experiment in a separate table rather than merging it into Table 3 because Table 3 is intended for controlled comparisons under our ImageNet-100 setting, whereas Table 11 follows the official LlamaGen setting. Combining them in one table could incorrectly suggest that all results are directly comparable.

---

> ### Author Response · Authors · 2026-05-01
> **Official Comparison with VAR and VQGAN**
>
> We thank the reviewer for this feedback and sincerely apologize for the lack of clarity. We agree that comparisons with established baselines are important for credibility, and we would like to clarify that we are not reluctant to include them. We show the comparison with official VAR, official VQGAN, and official LlamaGen tokenizer results under commonly reported tokenizer evaluation settings(**256×256 ImageNet-1k validation set**)[1] [2] [3].
>
> | Method                                | r-FID ↓ | PSNR ↑ | SSIM ↑ | Vendi ↑ | Euc. |   Cos. |
> | ------------------------------------- | ------: | -----: | -----: | ------: | ---: | -----: |
> | Official VQGAN[1]                     |   4.98* |      - |      - |    1.50 | 1.59 | 1.0000 |
> | LlamaGen[2] (follows VQGAN[1] design) |   2.19* |  20.79 |  0.675 |    5.29 | 1.42 | 0.9927 |
> | Ours                                  |    2.12 |  20.82 |  0.676 |    5.38 | 6.48 | 0.9956 |
> | VAR [3]                               |   0.90* |      - |      - |    5.61 | 1.11 | 1.0000 |
>
> \* indicates that the result is directly cited from the original paper or official GitHub repository.
>
> We would like to clarify that **the official VAR tokenizer achieves the strongest r-FID partly because it is trained on OpenImages(\~600M), which is substantially larger than ImageNet-1K(\~130M).** In addition, the official VAR tokenizer training code and detailed training recipe are not publicly released, making exact reproduction difficult. In our ImageNet-100 experiments, we used the XQGAN implementation as a practical and reproducible tokenizer backbone for VAR. However, matching the full ImageNet-1K-scale setting would require a very large training setup, e.g., a batch size of 1024 with 32 A100 GPUs or 128 Intel Habana Gaudi HPUs, which is beyond the resources available to us as an academic lab.
>
> We also attempted to run the official VQGAN repository. However, since the repository is several years old, **we encountered deadlocks and package conflicts under our hardware environment,** which we suspect are due to system compatibility issues. Moreover, **following the original training setting would require approximately 45.8 days on a single GPU, as reported in the original work, which is not feasible within the rebuttal timeline.**
>
> Therefore, instead of omitting official baselines, we include them as compared points and use the LlamaGen repository as a practical and reproducible training pipeline, **because it follows the VQGAN tokenizer design while providing a modern and usable training pipeline.** Under this setting, our method improves r-FID from 2.19 to 2.12 and also improves codebook coverage metrics, especially Vendi Score.
>
> We acknowledge this as a limitation of our current empirical evaluation. In the revised manuscript, we will explicitly discuss this limitation and clarify that we do not claim to outperform the official VAR tokenizer under its full-scale OpenImages training setting. Rather, our goal is to show that Deferred Quantization improves reconstruction and token coverage under matched and reproducible settings.
>
> To further improve transparency and reproducibility, we will release our implementation, training configurations, and evaluation scripts upon publication.
>
>
> [1] Esser, Patrick, Robin Rombach, and Bjorn Ommer. "Taming transformers for high-resolution image synthesis." *Proceedings of the IEEE/CVF conference on computer vision and pattern recognition*. 2021.
>
> [2] Sun P, Jiang Y, Chen S, et al. Autoregressive model beats diffusion: Llama for scalable image generation[J]. arXiv preprint arXiv:2406.06525, 2024.
>
> [3] Tian K, Jiang Y, Yuan Z, et al. Visual autoregressive modeling: Scalable image generation via next-scale prediction[J]. Advances in neural information processing systems, 2024, 37: 84839-84865.

---

> > ### Comment · Reviewer_pFXH · 2026-05-01
> >
> > I want to thank the authors for their prompt response and for the good additions.
> >
> > My additional questions:
> > 1. While Table 11 is presented, I don't see any explanation for it in the text (searching "table 11" or "llamagen" gives nothing in the main text).
> > 2. Is the LLamaGen w/ deferred quantization trained from scratch? Please describe its training process.
> > 3. For completeness, I encourage the authors to complete the missing PSNR and SSIM results for the added baselines.

---

> > > ### Author Response · Authors · 2026-05-01
> > >
> > > We thank the reviewer for the careful reading and helpful suggestions.
> > >
> > > > Q1: Table 11 is presented but not explained in the main text.
> > >
> > > We apologize for the late upload of revised manuscript. In the revised manuscript, we have moved this comparison into the main text as Table 7 under a new Section 6.4, "Comparison with Established Tokenizers," which provides a full discussion of the LlamaGen comparison and the official VQGAN/VAR baseline context.
> > >
> > > > Q2: Is LlamaGen w/ Deferred Quantization trained from scratch? Please describe its training process.
> > >
> > > Yes, the LlamaGen tokenizer with Deferred Quantization is trained entirely from scratch following the original LlamaGen tokenizer architecture and training recipe, with our two-stage Deferred Quantization protocol applied.
> > > Specifically, the training consists of:
> > >
> > > **Continuous Phase:** We first train the encoder-decoder as a continuous autoencoder on ImageNet-1K for around 5 epochs, corresponding to approximately 50,000 steps. During this phase, no vector quantization is used, and there is no codebook loss or commitment loss. The model is trained only with the reconstruction-related losses used by the LlamaGen tokenizer setting, including MSE, perceptual loss, and adversarial loss.
> > >
> > > **Discretization Phase:** After the continuous phase, we initialize the codebook using encoder embeddings from the pretrained continuous model, enable vector quantization, and continue training for 40 epochs. The tokenizer architecture, including the codebook size of 16,384 and latent dimension of 8, as well as the optimizer and other hyperparameters, follows the original LlamaGen tokenizer setting.
> > >
> > > Compared with the 40-epoch LlamaGen baseline, our method adds approximately 5 epochs of continuous autoencoder training. This phase is relatively lightweight because it does not involve nearest-codebook lookup over 16,384 entries, codebook loss, commitment loss, or codebook EMA updates. We also note that our ablation on the duration of the Continuous Phase (Table 9) suggests that the benefit appears early: on CIFAR-10, even a short continuous phase substantially improves codebook perplexity and reconstruction quality, with later epochs providing diminishing additional gains. This supports our interpretation that the main benefit comes from establishing a more dispersed encoder representation for codebook initialization, rather than merely from adding more training steps. We will add these training details to Section 6.4 to improve reproducibility.
> > >
> > >
> > > > Q3: For completeness, please complete the missing PSNR and SSIM results for the added baselines.
> > >
> > >
> > > We have computed and added PSNR and SSIM for the official VQGAN and VAR baseline using the publicly available checkpoint.

---

### Review · Reviewer_dQdy · 2026-03-25

**Summary Of Contributions:**

## summary
This paper identifies token representation shrinkage as one of the causes of mode collapse in VQ-VAE and proposes deferred quantization to address this issue. Specifically, deferred quantization is a method in which the codebook is initialized after pretraining an autoencoder without a quantizer. As a result, diversity is improved at both the embedding level and the generated image level, and performance gains are also confirmed in terms of FID and MSE.

## strength
- Comprehensive experiments demonstrate that deferred quantization outperforms conventional initialization based on an untrained encoder.
- The results in Figure 4 are interesting, showing that the proposed method is particularly effective when the number of tokens is increased.


## weakness
- The novelty of the proposed method may be limited. The method itself is based on pretraining an autoencoder before introducing the quantizer in VQ-VAE training. Therefore, the value of this paper largely depends on the analysis of the effects brought by this approach and the evaluation of downstream performance.
- The change in the encoder output distribution before and after training is not experimentally shown. Although this is mentioned in Figure 6, it would be better to include it in the main text. Figure 3 is purely conceptual and does not provide direct evidence supporting the claims of the paper.
- The results in Figure 2 are based on extremely small input dim and token dim, whereas the actual experiments are reported in the appendix to use token dimensions of 32 or 64. It is unclear whether the trends shown in Figure 2 are also observed in more realistic settings with larger input and token dimensions.
- There are missing descriptions regarding the training procedure and analysis methods, which reduces the clarity of the paper (specific points are detailed in the requested changes). Although some of these details are provided in the appendix, it would improve clarity to include them in the main text.
- To demonstrate the effectiveness of the proposed two-stage method, the paper should also include a comparison with a setup where all 200 epochs are trained without quantization, followed by post-hoc discretization using k-means.
- The paper lacks detailed descriptions regarding normalization when using metrics in the embedding space, such as Euclidean distance. For example, in Table 2, Euc. is reported to increase with deferred quantization; however, if the comparison is made without proper normalization, this improvement may simply reflect a global scaling of the embedding space. It is therefore necessary to control for the overall scale and demonstrate that the pairwise distances between points increase under a consistent normalization scheme.

**Audience:**

Yes

**Audience Explanation:**

Tokenization of images and audio has recently attracted significant attention in multimodal research. The identification of representation shrinkage as a cause of mode collapse, as well as the proposed training strategy to mitigate it, could provide useful insights for the research community.

**Broader Impact Concerns:**

There are no significant broader impact concerns.

**Claims And Evidence:**

Yes

**Claims Explanation:**

While there are several concerns, the paper overall demonstrates that deferred quantization alleviates representation shrinkage and contributes to improved downstream performance, supported by both analyses in the embedding space and evaluations on downstream tasks.

**Requested Changes:**

## Major changes
- While the intended message of Figure 2 can be understood, the description lacks sufficient detail about what specific values are being plotted, making it difficult to assess the scientific validity. In particular, the difference between the top row (Reconstruction) and the bottom row (Token Distribution) is unclear.
- To justify the claim that pretraining makes the encoder outputs more well-distributed, the authors should include Figure 6 in the main text or replace Figure 3 with plots based on actual data rather than a conceptual illustration.
- The term "semantic embeddings" is used repeatedly, but it is not clearly defined. From the appendix, it can be inferred that these correspond to k-means centroids of the encoder outputs; however, this should be explicitly stated in the main text.
- In Section 3.2, the paper states "we define the VQ-VAE as …", but it is unclear whether this differs from the original formulation in van den Oord et al. (2017). If any modifications have been made, they should be clearly specified; at the very least, the original work should be properly cited.
- The number of epochs without the quantizer and those used for subsequent fine-tuning are crucial aspects of the proposed method, and should therefore be included in the main content rather than in the appendix.
- Could you provide additional details on the use of distance metrics in the embedding space (e.g., Euclidean distance), particularly regarding whether normalization is applied and how it is performed? If the overall scale of the embedding space differs, comparisons of distances may become difficult to interpret without appropriate normalization (e.g., L2 normalization).

## Minor changes
- To demonstrate the effectiveness of the proposed two-stage training protocol, I recommend including a comparison with an alternative approach in which all training steps are performed without a quantizer, followed by post-hoc discretization using k-means. While the current experiments demonstrate the effectiveness of the geometric phase, they do not yet clearly establish the benefit of the discretization phase, where the quantizer is jointly trained with the encoder-decoder. In addition, the number of epochs for the geometric phase is fixed at 100 in the current setup; an ablation study varying this from 0 to 200 epochs would also be informative.
- Why do Figures 6(a) and 6(b) show values with different orders of magnitude on both the x- and y-axes?
- If possible, Figure 2 should be plotted using the same dimensionality as in the actual experiments. This would help address the question of whether the observed issue could be resolved simply by increasing the dimensionality, even without deferred quantization.
- While the paper shows that deferred quantization alleviates representation shrinkage, it seems plausible that similar effects could be achieved without pretraining, for example, by using initialization schemes that encourage larger variance or by applying appropriate normalization to make the representations more well-distributed. Including comparisons with such alternatives would help strengthen the claim of the proposed method’s effectiveness.

---

> ### Author Response · Authors · 2026-04-18
> **Major Changes 1 - 5**
>
> > Major C1: While the intended message of Figure 2 can be understood, the description lacks sufficient detail about what specific values are being plotted, making it difficult to assess the scientific validity. In particular, the difference between the top row (Reconstruction) and the bottom row (Token Distribution) is unclear.
> >
>
> We thank the reviewer for this feedback. We agree that the original Figure 2 lacked sufficient detail regarding what was being plotted, and the distinction between the two rows was unclear. We have revised the figure to address these concerns.
>
> The updated Figure 2 now presents two complementary settings at different dimensionalities:
>
> - **Subfigure (a)** (Input dim = 2, Token dim = 1): Since the latent space is 1-dimensional, we directly visualize the embedding and token distributions via kernel density estimation (KDE), requiring no dimensionality reduction. The left column shows data-space reconstructions; the right column shows the latent-space distributions.
> - **Subfigure (b)** (Input dim = 48, Token dim = 32): For the higher-dimensional case, we use t-SNE to project both the data space and the latent space into 2D for visualization.
>
> In both figures, the top row shows results **without** deferred quantization and the bottom row shows results **with** deferred quantization. The color coding is consistent throughout: light blue = original input, dark blue = reconstruction, light pink = encoder embeddings, dark pink = quantized tokens.
>
> The revised figure makes the following observation clear: without deferred quantization, tokens collapse into a narrow region of the latent space (visible as concentrated pink clusters), leading to mode-collapsed reconstructions in data space. With deferred quantization, tokens spread to cover the embedding distribution, and reconstructions faithfully recover all input modes. This pattern is consistent across both dimensionality settings. We have also updated the caption to provide a precise description of all plotted quantities.
>
> > Major C2: To justify the claim that pretraining makes the encoder outputs more well-distributed, the authors should include Figure 6 in the main text or replace Figure 3 with plots based on actual data rather than a conceptual illustration.
> >
>
> Thanks for your suggestion for clarity. We have moved Figure 6 to the main text in the revised manuscript.
>
> > Major C3: The term "semantic embeddings" is used repeatedly, but it is not clearly defined. From the appendix, it can be inferred that these correspond to k-means centroids of the encoder outputs; however, this should be explicitly stated in the main text.
> >
>
> We thank the reviewer for this helpful suggestion. We agree that the term "semantic embeddings" was not clearly defined. In our method, this term refers to the encoder outputs obtained after continuous autoencoder pretraining, before quantization is enabled. In the revised manuscript, we have replaced "semantic embeddings" with the more precise term "pretrained encoder embeddings" throughout.
>
> > Major C 4: In Section 3.2, the paper states "we define the VQ-VAE as …", but it is unclear whether this differs from the original formulation in van den Oord et al. (2017). If any modifications have been made, they should be clearly specified; at the very least, the original work should be properly cited.
> >
>
> We thank the reviewer for raising this question. Our formulation is identical to the original VQ-VAE of van den Oord et al. (2017) except for notation. The phrasing "we define the VQ-VAE as..." was meant to introduce notation, not to propose a new formulation, and we apologize for the ambiguity.
>
> In the revised version, we have added an explicit citation and modified the expression "Following standard VQ-VAE (Van Den Oord et al., 2017), we adopt the formulation …"
>
> > Major C 5: The number of epochs without the quantizer and those used for subsequent fine-tuning are crucial aspects of the proposed method, and should therefore be included in the main content rather than in the appendix.
> >
>
> We thank the reviewer for this suggestion. We have moved the training schedule details to **Section 6.1** Experiment setup in the revised version.

---

> ### Author Response · Authors · 2026-04-18
> **Major Changes 6**
>
> > Major C6 Could you provide additional details on the use of distance metrics in the embedding space (e.g., Euclidean distance), particularly regarding whether normalization is applied and how it is performed? If the overall scale of the embedding space differs, comparisons of distances may become difficult to interpret without appropriate normalization (e.g., L2 normalization).
> >
>
> We thank the reviewer for this methodological point. We provide the computation details below and add two scale-invariant diversity metrics to address the normalization concern.
>
> **(1) Euclidean distance.** Our reported Euclidean distance is the mean pairwise L2 distance across all codes:
>
> $D_{\text{Euc}} = \frac{2}{S(S-1)} \sum_{i < j} \|t_i - t_j\|_2.$
>
> **(2) Cosine distance.** To remove magnitude sensitivity as the reviewer suggests, we add the mean pairwise cosine distance(normalized the code by L2 normalization):
>
> $D_{\text{cos}} = \frac{2}{S(S-1)} \sum_{i < j} \left(1 - \frac{t_i^\top t_j}{\|t_i\| \, \|t_j\|}\right).$
>
> **(3) Vendi Score.** We additionally report the Vendi Score (Friedman & Dieng, TMLR 2023) to capture overall codebook diversity. We follow them to construct the similarity matrix via the inner-product kernel $K_{ij} = t_i^\top t_j$ and compute
>
> $\text{VS}(T) = \exp\left(-\sum_{i} \lambda_i \log \lambda_i\right),$
>
> where $\lambda_i$ are the eigenvalues of $K / \mathrm{tr}(K)$.
>
> **(4)** The table reports all three metrics of VAR tokenizer on ImageNet-100. Improvements under Deferred Quantization hold for both scale-invariant metrics cosine distance and Vendi score), confirming that the Deferred Quantization improves shrinkage and enables a diverse codebook distribution. Full results across all settings are available in the revised manuscript.
>
> | VAR Tokenizer | r-FID ↓ | MSE ↓(×10^-3) | LPIPS ↓(×10^-2) | Euc. ↑ | Cos. ↑ | Vendi. ↑ | Perp. ↑ |
> | --- | --- | --- | --- | --- | --- | --- | --- |
> | w/o Deferred | 5.39 | 2.60 | 1.85 | 1.19 | 0.64 | 2.19 | 2801.88 |
> | + SimVQ | 5.52 | 2.50 | 1.78 | 0.80 | 0.96 | 2.07 | 6920.23 |
> | w/ Deferred | 5.04 | 2.22 | 1.63 | **7.56** | 0.97 | 2.26 | 7044.51 |
> | + SimVQ | **4.93** | **2.17** | **1.59** | 0.93 | **1.00** | **3.47** | **8222.83** |

---

> ### Author Response · Authors · 2026-04-18
> **Minor Changes 1 - 2**
>
> > Minor C 1: To demonstrate the effectiveness of the proposed two-stage training protocol …
> >
>
> We thanks for reviewer’s suggestion. And conduct experiments on
>
> We thank the reviewer for this insightful suggestion. We agree that comparing with post-hoc k-means discretization is important to establish the benefit of the Discretization Phase. We conduct this experiment on CIFAR-10 with a codebook size of 8192 using three configurations:
>
> 1. **Baseline VQ-VAE**: standard VQ-VAE trained for 300 epochs.
> 2. **Deferred Quantization**: 150 epochs geometric phase(continuous phase) + 150 epochs discretization phase (total 300 epochs).
> 3. **Post-hoc k-means**: autoencoder trained for 300 epochs without quantization, followed by k-means discretization on the encoder outputs (no joint VQ fine-tuning).
>
> Table 2. Ablation for discretization phase
>
> | Method | MSE↓(×10^-3) | Perp. ↑ | Vendi.↑ |
> | --- | --- | --- | --- |
> | Baseline VQ-VAE (300 ep) | 9.6 | 2018.63 | 1.66 |
> | Post-hoc k-means (300 ep AE) | 21.0 | 2724.57 | 2.04 |
> | Deferred Quant. (150 + 150 ep) | 9.2 | 3104.97 | 1.95 |
>
> Deferred Quantization outperforms post-hoc k-means, demonstrating that the discretization phase (where the quantizer is jointly trained with the encoder-decoder) provides additional gains beyond what static k-means discretization can achieve. Joint fine-tuning allows the encoder, decoder, and codebook to co-adapt, closing the gap between continuous and discrete representations and yielding better reconstruction quality.
>
> Additionally, we conducted a simple ablation to study the effect of pretraining quality on Cifar10. We fixed the discretization phase to 150 epochs and used a different Geometric phase. Our results are shown below. We find that **even a short geometric phase (e.g., 10 epochs)** is sufficient to improve VQ-VAE performance. Beyond **20 epochs**, continued autoencoder training yields only marginal gains in VQ-VAE performance.
>
> This suggests that the benefits of pretraining plateau after a certain point, and moderate geometric phase is already sufficient to mitigate token representation shrinkage.
>
> Table 1. Validation loss(MSE ×10^-3) for AE.
>
> | Epoch | 0 | 10 | 20 | 40 | 60 | 80 | 100 |
> | --- | --- | --- | --- | --- | --- | --- | --- |
> | Val loss | 24.4 | 4.41 | 2.63 | 1.43 | 1.09 | 0.76 | 0.63 |
>
> Table 2.  Performance of VQVAE initialized by different AE.
>
> | AE Epoch | 0 | 10 | 20 | 40 | 60 | 80 | 100 |
> | --- | --- | --- | --- | --- | --- | --- | --- |
> | MSE (×10^-3) | 9.65 | 9.32 | 9.23 | 9.30 | 9.23 | 9.20 | 9.38 |
> | Perplexity | 2018.63 | 3127.93 | 3143.12 | 3126.99 | 3089.81 | 3107.58 | 3039.83 |
>
> > Minor C 2: Why do Figures 6(a) and 6(b) show values with different orders of magnitude on both the x- and y-axes?
> >
>
> Thank you for the question. The trained and untrained encoders naturally occupy different x-ranges: the untrained encoder (red) is highly concentrated within the narrow interval [0.0, 0.4], while the trained encoder (blue) spreads over a much wider range of approximately [-4, 4]. In addition, we plot the distributions using Kernel Density Estimation (KDE), where the y-axis represents normalized probability density rather than counts. Therefore, the peak height depends on how concentrated the distribution is, and different supports naturally lead to different y-axis magnitudes.

---

> ### Author Response · Authors · 2026-04-18
> **Minor Changes 3 - 4**
>
> > Minor C 3: If possible, Figure 2 should be plotted using the same dimensionality as in the actual experiments. This would help address the question of whether the observed issue could be resolved simply by increasing the dimensionality, even without deferred quantization.
> >
>
> We thank the reviewer for this helpful suggestion. We have added an additional experiment to Figure 2 using a higher-dimensional setting (input dimension = 48, token dimension = 32), which is closer to the dimensionality used in our actual experiments. The result shows a consistent trend: the coverage failure persists without deferred quantization, suggesting it is not resolved simply by increasing dimensionality.
>
> > Minor C 4: While the paper shows that deferred quantization alleviates representation shrinkage, it seems plausible that similar effects could be achieved without pretraining, for example, by using initialization schemes that encourage larger variance or by applying appropriate normalization to make the representations more well-distributed. Including comparisons with such alternatives would help strengthen the claim of the proposed method’s effectiveness.
> >
>
> We thank the reviewer for this constructive suggestion. To address this concern, we conducted additional experiments comparing deferred quantization with variance-scaled codebook initialization, where the codebook vectors are drawn from Gaussian distributions with varying standard deviations (σ ∈ {1.0, 2.0, 3.0, 10.0}) and quantization is applied from the beginning of training (i.e., no deferred pretraining). The results are summarized in Table 3.
>
> *Table 3. Comparison of deferred quantization with variance-scaled codebook initialization.*
>
> | Method | MSE ↓(×10^-3) | Perplexity ↑ | Vendi ↑ |
> | --- | --- | --- | --- |
> | Gaussian Init (σ=1.0) | 9.6 | 1889.0 | 1.579 |
> | Gaussian Init (σ=2.0) | 9.8 | 1777.5 | 1.552 |
> | Gaussian Init (σ=3.0) | 9.8 | 1930.0 | 1.557 |
> | Gaussian Init (σ=10.0) | 9.6 | 2207.4 | 1.577 |
> | **Deferred Quantization (Ours)** | **8.5** | **4079.4** | **1.953** |
>
> As shown in Table 3, varying the initialization variance has only a marginal effect: both the MSE and the Vendi score remain largely unchanged across different values of σ. This supports our central argument that the key bottleneck is not the initial spread of the codebook vectors, but whether the codebook is initialized from meaningful embeddings. At the same time, we agree that this is a promising direction, and we plan to investigate such initialization strategies more systematically in future work.

---

> > ### Comment · Reviewer_dQdy · 2026-04-28
> > **On the contribution of the paper**
> >
> > Thank you for addressing the points raised earlier. My main concerns regarding the initial submission have largely been resolved. However, I have several questions about the revised manuscript.
> >
> > A paragraph discussing related approaches has been added at the end of the Related Work section. From this, I get the impression that Deferred Quantization itself may not be novel, and that the primary contribution of this paper is the finding that its effectiveness stems from mitigating Token Coverage Failure. Is this an accurate interpretation?
> > If so, I feel that the discussion of the methodological differences between Deferred Quantization and approaches such as TokenBridge and ReVQ seems incomplete. If these methods are essentially identical, this should be stated explicitly. Otherwise, any differences should be clearly articulated.
> >
> > (although this is a relatively minor point) Additionally, the paper shows that introducing Deferred Quantization leads to both the mitigation of Token Coverage Failure and improved performance. However, this only demonstrates that these two effects occur together, and does not necessarily mean that Token Coverage Failure is the bottleneck responsible for the performance degradation in vanilla VQ-VAE (although this appears intuitively plausible).
> > I had expected that, if even a more naive approach to mitigating Token Coverage Failure were to yield some degree of improvement, this would provide supporting evidence for this claim. However, the results in Table 9 do not appear to align with this expectation.
> > If Deferred Quantization itself is intended as a novel methodological contribution, demonstrating improved performance alone may be sufficient, and such a level of rigor may not be necessary. However, if the prior work has already shown that Deferred Quantization can improve performance and this paper is positioned more as an analytical study, it could be beneficial to also consider these aspects.

---

> ### Author Response · Authors · 2026-04-28
>
> **About Related Work Section:** We thank the reviewer for this question. Both TokenBridge, ReVQ, and our method share the high-level idea of decoupling continuous learning from discretization. However, both TokenBridge and ReVQ focus on transferring the reconstruction quality of a pretrained VAE to a discrete tokenizer with minimal degradation. In contrast, our work focuses on diagnosing why early quantization induces Token Coverage Failure and how this propagates to reduced generative diversity downstream. And a key methodological difference is that both TokenBridge and ReVQ freeze the encoder/decoder during discretization. In contrast, our method jointly fine-tunes all components. We will revise the Related Work to clarify these distinctions.
>
>
> **Why variance-scaled codebook initialization failed:** We thank the reviewer for this observation. We would like to clarify why variance-scaled initialization does not appear to mitigate Token Coverage Failure. We believe the key factor may not lie in the initial spread of codebook vectors, but rather in the alignment between the codebook and the encoder's learned representation structure. Randomly dispersed codebook vectors bear no relation to the encoder's initial output distribution, nor to the embeddings it will ultimately learn; once training begins, nearest-neighbor assignment tends to pull them back toward the untrained encoder's narrow output distribution, erasing the initial spread. This interpretation is indicated by the Vendi Score in Table 9: across all variance scales (σ = 1.0 to 10.0), the Vendi Score remains nearly unchanged at around 1.55–1.58, suggesting that the intrinsic codebook diversity is largely unaffected by initialization variance. In contrast, Deferred Quantization raises the Vendi Score to 1.953, indicating a more meaningful improvement in codebook diversity.

---

> > ### Comment · Reviewer_dQdy · 2026-04-28
> > **Official Comment by Reviewer dQdy**
> >
> > Thank you for the clarification that, in TokenBridge and ReVQ, the encoder and decoder are frozen during discretization. I think that this constitutes a distinguishing feature of Deferred Quantization, and it is sufficient to argue that the proposed method differs methodologically from these approaches. It would be helpful to include this point in the manuscript.
> >
> > I also appreciate the explanation regarding the interpretation of the results in Table 9. As I understand it, diversity at initialization is only a necessary condition for preventing Token Coverage Failure, and in addition, it is important that the distributions of the encoder outputs and the codebook are reasonably well aligned.
> > It might be worthwhile to investigate whether there are any established methods for increasing the variance of the outputs of an untrained encoder through its initialization, rather than merely modifying the initialization of the codebook. If this setting is effective even with simple k-means codebook initialization, it could more clearly demonstrate the importance of preventing Token Coverage Failure as well as the effectiveness of Deferred Quantization. (If there do not appear to be any such established methods, please feel free to ignore this comment.)

---

> > > ### Author Response · Authors · 2026-04-28
> > >
> > > Thank you for your positive feedback. We have added this distinction to the revised Related Work section.
> > >
> > > Regarding encoder initialization: based on our investigation, and to the best of our knowledge, we have not found an established method that is specifically designed to increase the variance of an untrained encoder’s outputs. Standard initialization schemes, such as Xavier initialization[1], Kaiming initialization[2], and LSUV(Layer-Sequential Unit-Variance) initialization[3], primarily aim to stabilize signal propagation and gradient flow during training, rather than to induce a well-dispersed output space before learning.
> > >
> > > [1] Glorot, Xavier, and Yoshua Bengio. "Understanding the difficulty of training deep feedforward neural networks." Proceedings of the thirteenth international conference on artificial intelligence and statistics. JMLR Workshop and Conference Proceedings, 2010.
> > >
> > > [2] He, Kaiming, et al. "Delving deep into rectifiers: Surpassing human-level performance on imagenet classification." Proceedings of the IEEE international conference on computer vision. 2015.
> > >
> > > [3] Mishkin, Dmytro, and Jiri Matas. "All you need is a good init." arXiv preprint arXiv:1511.06422 (2015).

---

> ### Comment · Reviewer_dQdy · 2026-04-29
> **Official Comment by Reviewer dQdy**
>
> Thank you for the clarification.
> Could you upload the revised version, or is it only possible to make changes after the final results are released? If it can be uploaded now, I will prepare the official recommendation based on it.

---

> > ### Author Response · Authors · 2026-04-29
> >
> > Thanks for your reminder. We have uploaded the revised version.
> >
> > Modifications
> > - We point out the distinction in Sec 2
> > - We add interpretation about Table 9 in Sec 6.4(page 13)

---

> > > ### Comment · Reviewer_dQdy · 2026-04-30
> > > **Official Comment by Reviewer dQdy**
> > >
> > > The explanation of the results of Table 9 is quite cluttered. There appears to be substantial overlap between the paragraph beginning with "We further examine whether …" and the preceding discussion. This makes the section very difficult to read. Please review and revise accordingly.

---

> > > > ### Author Response · Authors · 2026-04-30
> > > >
> > > > We thank the reviewer for pointing this out. We have revised and streamlined the discussion of Table 9 to remove the redundancy. The updated version has been uploaded. We welcome any further feedback and are willing to revise accordingly.

---

### Review · Reviewer_nuZ6 · 2026-04-05

**Summary Of Contributions:**

This paper proposes to delay the learning of VQ after some training of the network. The observation is that the initial distributions of the embedding values change significantly after several training iterations. Initializing the codebook with embeddings from an undertrained encoder can make it difficult to capture the moments of distribution shift after training. The approach is simple and improves the vanilla VQ optimization approach. Instead of optimizing the VQ layer along with other parts of the network, the paper initiates the training of it after the network is properly trained.

**Audience:**

Yes

**Audience Explanation:**

The optimization of vector quantization has been an important problem in neural compression. The technique has also been widely adopted to quantize continuous signals such as speech and images to interact with large language models. The topic therefore should be of interest to TMLR's audience.

**Broader Impact Concerns:**

I do not see any concerns.

**Claims And Evidence:**

No

**Claims Explanation:**

- I do not see any theoretical findings from 4.2. The bound in eq (6) does not say anything new. There is also no theoretical connections or proofs that: "shrinkage can simultaneously reduce diversity and increase distortion"

- In the abstract, it is unclear what is the latent manifold or why the latent space is a manifold. There is no follow-up explanation of the term in the paper. Same issue occurs in Section 5, the use of representation manifold:
"This protocol encourages the codebook to adapt to an already-formed representation manifold, mitigating shrinkage and improving coverage."

- It is unclear what are semantic embeddings used to initialize the codebook. There's also not evidence showing that the embeddings used to learn the codebook learn semantic distinctions.

- Could you please explain what is Geometric Phase? What's the difference between Geometric Phase v.s. training a normal encoder-decoder network? In particular how it leads to a structured, dispersed embedding space (or semantic space mentioned above). The current experiments does not seem to support this claim.

- The paper fails to cite relavent work that directly address the same issue [1]. The paper does not compare to approaches that aim to improve the optimization of the VQ layer, e.g., [1], [2].

- I’m not sure "shrinkage" is the right term to describe the problem. The codebook distribution doesn’t actually shrink. Instead, it is initialized using the network’s embeddings at the beginning, but over time this leads to a mismatch between the codebook distribution and the embedding distribution. So the core issue is a distribution mismatch.

- The paper fails to cite relavent work that directly address the same issue [1]. The paper does not compare to approaches that aim to improve the optimization of the VQ layer, e.g., [1], [2].

- In 3.2 Vector Quatization, it is unclear what are the objectives used for the VQ training in the paper, there's no appropriate reference provided. Why this specific choice of baseline instead of other approaches that are shown to be less prone to mode collapse?

[1] Straightening Out the Straight-Through Estimator: Overcoming Optimization Challenges in Vector Quantized Networks

[2] SoundStream: An End-to-End Neural Audio Codec

**Requested Changes:**

- The submission did not sufficiently cover prior work addressing optimization challenges in VQ layers. Please include comparisons with methods that explicitly improve VQ optimization as suggested above.
- The term shrinkage may be misleading. Consider rename it. The codebook distribution does not contract over time; rather, it is initialized from early encoder embeddings, which later diverge from the learned embedding space.
- Section 4.2 does not add new insights. Consider remove it or properly formulate the equations to support your claims on the relationship between codebook usage and reconstruction.
- Clarify the term "latent manifold",  "semantic embeddings", "Geometric Phase".
- Clarify the training objectives used for vector quantization.
- Specify when does the codebook training start after the training of encode-decoder. What's the loss curves of this two-stage training compared to other approaches.

---

> ### Author Response · Authors · 2026-04-18
> **Changes 1**
>
> > C 1:  The submission did not sufficiently cover prior work addressing optimization challenges in VQ layers.
> >
>
> We thank the reviewer for this suggestion. We have conducted experiments on CIFAR-10 comparing Deferred Quantization with the methods referenced by the reviewer, across codebook sizes ranging from 1,024 to 32,768.
>
> **Regarding [1] (Huh et al., "Straightening Out the Straight-Through Estimator").** This paper proposes 3 techniques: Affine Parameterization, Alternated Optimization, and Synchronous Update. We implemented all three using the official tool library `vqtorch` from the [1] . However, Alternated Optimization consistently led to strong code collapse in different settings(Training losses and perplexity available in Appendix A 3 Figure 8). We assume this may be due to differences in library dependencies and hardware configurations that have evolved since the original publication; we are currently trying to contact the authors to resolve this.
>
> For the Affine + Sync combination, we used the default parameters provided in `vqtorch` (affine_lr = 2, sync_nu = 2). We additionally tries to do hyperparameter tuning on the 1,024-code setting for affine parameterization and Synchronous Update, respectively, yielding affine_lr = 5 and sync_nu = 0.1, and report results for both configurations.
>
> **Regarding [2] (Zeghidour et al., SoundStream).** We follow the codebook optimization strategy proposed in SoundStream: the centroids are initialized by running k-means on the training batch, and dead tokens are subsequently replaced during training under a least-recently-used (LRU) policy.
>
> **Results are shown in the tables below.**
>
> **Table: MSE ↓(×10^-3) on CIFAR-10 across codebook sizes.**
>
> | Method | 1024 | 4096 | 8192 | 16384 | 32768 |
> | --- | --- | --- | --- | --- | --- |
> | VQVAE | 12.39 | 10.99 | 10.15 | 10.00 | 9.80 |
> | VQVAE + EMA | **11.49** | 10.15 | 9.65 | 9.77 | 9.72 |
> | Affine + Sync (default) * | 247.67* | 13.26 | 11.65 | 12.15 | 11.93 |
> | Affine (affine_lr=5) | 12.06 | **9.86** | 10.23 | 10.17 | 9.43 |
> | Sync (sync_nu=0.1) |12.17|11.53|11.43|11.10|10.98|
> | LRU + K-means (SoundStream) | 12.24 | 10.00 | **9.20** | 8.62 | 8.32 |
> | Deferred Quant. (Ours) | 12.32 | 10.26 | 9.35 | 8.75 | 8.15 |
> | Deferred Quant. + EMA (Ours) | 12.49 | 10.13 | 9.23 | **8.46** | **8.00** |
>
> *: Affine + Sync (default) training failure at codebook size 1,024.(We provide the loss in Appendix A.3 Figure 7)
>
> **Table: Perplexity on CIFAR-10 across codebook sizes.**
>
> | Method | 1024 | 4096 | 8192 | 16384 | 32768 |
> | --- | --- | --- | --- | --- | --- |
> | VQVAE | 753.67 | 1481.82 | 2489.56 | 2235.81 | 2532.43 |
> | VQVAE + EMA | 750.94 | 1509.93 | 2018.63 | 1837.66 | 1974.73 |
> | Affine + Sync (default) * | 1.0* | 460.33 | 859.37 | 738.97 | 819.92 |
> | Affine (affine_lr=5) | 783.65 | 2239.34 | 2011.03 | 1941.82 | 2999.24 |
> | Sync (sync_nu=0.1) |726.15|1098.52|988.92|1110.11|1306.09|
> | LRU + K-means (SoundStream) | **786.35** | **2341.85** | **3777.72** | **5188.40** | **6030.98** |
> | Deferred Quant. (Ours) | 637.82 | 1786.02 | 2712.89 | 3671.48 | 4548.58 |
> | Deferred Quant. + EMA (Ours) | 712.86 | 2082.09 | 3104.97 | 4079.40 | 4839.72 |
>
> **Table: Vendi Score on CIFAR-10 across codebook sizes.**
>
> | Method | 1024 | 4096 | 8192 | 16384 | 32768 |
> | --- | --- | --- | --- | --- | --- |
> | VQVAE | 1.89 | 1.71 | 1.66 | 1.55 | 1.50 |
> | VQVAE + EMA | **2.03** | 1.80 | 1.70 | 1.58 | 1.52 |
> | Affine + Sync (default) * | 1.03 | 1.53 | 1.58 | 1.50 | 1.47 |
> | Affine (affine_lr=5) | 1.86 | 1.82 | 1.63 | 1.53 | 1.53 |
> | Sync (sync_nu=0.1) |1.92|1.65|1.53|1.50|1.47|
> | LRU + K-means (SoundStream) | 1.87 | 1.83 | 1.81 | 1.77 | 1.75 |
> | Deferred Quant. (Ours) | 1.89 | 1.96 | 1.93 | **1.97** | 1.90 |
> | Deferred Quant. + EMA (Ours) | 1.98 | **1.97** | **1.95** | 1.95 | **1.99** |
>
> **Deferred Quantization scales favorably with codebook size.** At small codebook sizes (1024), Deferred Quantization performs inferior to other methods. However, as the codebook size increases, its reconstruction quality improves substantially, becoming comparable to LRU + K-means (SoundStream) at 8,192 codes and surpassing it at larger sizes.
>
> **Deferred Quantization maintains consistently high Vendi Scores.** Regardless of codebook size, both Deferred Quantization and Deferred Quantization + EMA maintain Vendi Scores around 1.90--1.99,  higher than all other methods. This indicates that our approach preserves greater latent space diversity across codebook sizes
>
> [1] Huh, Minyoung, et al. "Straightening out the straight-through estimator: Overcoming optimization challenges in vector quantized networks." *International Conference on Machine Learning*. PMLR, 2023.
>
> [2] Zeghidour, Neil, et al. "Soundstream: An end-to-end neural audio codec." *IEEE/ACM Transactions on Audio, Speech, and Language Processing* 30 (2021): 495-507.
>
> [3] Van Den Oord, Aaron, and Oriol Vinyals. "Neural discrete representation learning." *Advances in neural information processing systems* 30 (2017).

---

> > ### Author Response · Authors · 2026-04-18
> > **Changes 2 - 4**
> >
> > > C 2:  The term shrinkage may be misleading. Consider rename it. The codebook distribution does not contract over time; rather, it is initialized from early encoder embeddings, which later diverge from the learned embedding space.
> >
> > We thank the reviewer for pointing out this ambiguity. The problem that we observe can be stated as follows: doing early quantisation results in a codebook that “underrepresents” the continuous embeddings (i.e. the encoder’s output space). A better word for this “underrepresentation” is coverage. That’s why we have renamed the phenomenon we observed from “token representation shrinkage” to “token coverage failure” and changed the manuscript accordingly. We hope it conveys our idea in a more accurate manner.
> >
> > > C 3:  Section 4.2 does not add new insights. Consider remove it or properly formulate the equations to support your claims on the relationship between codebook usage and reconstruction.
> >
> > We thank the reviewer for this suggestion. We agree that Section 4.2 does not add substantive new insights, and we have removed it in the revised version.
> >
> > > C 4:  Clarify the term "latent manifold", "semantic embeddings", "Geometric Phase".
> >
> > We thank the reviewer for pointing out these terminological ambiguities. We have revised the manuscript to clarify all three terms as follows.
> >
> > **"Latent manifold"**
> > We agree that these terms were used loosely without formal justification. In the revised manuscript, we replace "latent manifold" with "latent space" respectively, throughout the abstract, Section 5, and other occurrences.
> >
> > **"Semantic embeddings."**
> >
> > We agree that the term "semantic embeddings" was imprecise and may suggest high-level semantic supervision, which is not what we intend. In the revised manuscript, we replace this term with the more precise description "pretrained encoder embeddings," i.e., encoder outputs obtained after continuous autoencoder pretraining.
> > Our claim is not that these embeddings encode semantics in the contrastive or supervised-learning sense. Rather, our point is: compared with outputs from a randomly initialized encoder, pretrained encoder embeddings provide a more stable and better-spread representation distribution for codebook initialization. This improves the initial coverage of the encoder embedding space when quantization is enabled.
> >
> > **"Geometric Phase."**
> > We also remove the term "Geometric Phase" and instead describe the method as a two-stage protocol: (Stage 1 Continuous Phase): continuous pretraining of a standard autoencoder, followed by (Stage 2 Discretization Phase): Initialize the codebook using pretrained encoder embeddings from the pretrained encoder, then enable vector quantization and continue training under the VQ objective.

---

> ### Author Response · Authors · 2026-04-18
> **Changes 5-6**
>
> > C 5:  Clarify the training objectives used for vector quantization.
> >
>
> In the revised version, Section 3.2 now explicitly states the full training objective. Following the original VQ-VAE [1], our overall loss is:
>
> $\mathcal{L} = \mathcal{L}{\text{recon}} + \mathcal{L}{\text{codebook}} + \beta \, \mathcal{L}_{\text{commit}},$
>
> where the codebook loss
>
> $\mathcal{L}{\text{codebook}} = || sg[z_j] - \hat{z}_j ||_2^2$
>
> encourages codebook entries to move towards the encoder outputs, the commitment loss
>
>  $\mathcal{L}{\text{commit}} = || z_j - sg[\hat{z}_j] ||_2^2$
>
>  encourages encoder outputs to stay close to the chosen codebook entries, and $\mathrm{sg}[\cdot]$ denotes the stop-gradient operator.
>
> The reconstruction loss  $\mathcal{L}{\text{recon}}$ is composed of MSE loss,
> perceptual loss [2],
> and adversarial loss [3],
> with the specific combination depending on the experimental setting.
>
> [1] Van Den Oord, Aaron, and Oriol Vinyals. "Neural discrete representation learning." *Advances in neural information processing systems* 30 (2017).
>
> [2] Zhang, Richard, et al. "The unreasonable effectiveness of deep features as a perceptual metric." *Proceedings of the IEEE conference on computer vision and pattern recognition*. 2018.
>
> [3] Esser, Patrick, Robin Rombach, and Bjorn Ommer. "Taming transformers for high-resolution image synthesis." *Proceedings of the IEEE/CVF conference on computer vision and pattern recognition*. 2021.
>
> > C 6: Specify when does the codebook training start after the training of encode-decoder. What's the loss curves of this two-stage training compared to other approaches.
> >
>
> When codebook training starts (Continuous Phase → Discretization Phase):
>
> - **Synthetic dataset:** The autoencoder is trained for 100 epochs during the Continuous Phase; quantization (codebook training) is then enabled for an additional 100 epochs during the Discretization Phase.
> - **CIFAR-10:** The encoder-decoder is trained for 50 epochs in the Continuous Phase, followed by 250 epochs with quantization enabled in the Discretization Phase. For Deferred Quantization with EMA-based codebook updates, we use 150 + 150 epochs for the two phases.
> - **ImageNet-100 and ODIR:** The encoder-decoder is first trained for 200 epochs in the Continuous Phase. Once quantization is enabled, both our method and all baselines are trained with a ReduceLROnPlateau scheduler (patience = 20): the learning rate is halved whenever the validation loss fails to improve for 20 epochs, decaying from 1×10^−4 to 1×10^−7. Training terminates via early stopping once no further improvement is observed at the minimum learning rate.
>
> We have added the loss curves in Appendix A.3 Figure 6.

---

### Decision · Action_Editor_vhwh · 2026-05-31

**Recommendation:** Reject

**Additional Comments:**

Overall, the reviewers feedback are valuable, and I highly encourage the authors use them to plan the next revision of the paper.

**Audience:**

Yes

**Audience Explanation:**

Quantizing vectors is a critical tool and is widely used in text, vision, and speech tasks. An improvement along this direction would be of interest to many and would have a good impact.

**Claims And Evidence:**

No

**Claims Explanation:**

Reviewer nuZ6 finds the claims unconvincing. The problem is not a lack of evidence, but a lack of precision in the statements and their utility. This explains why reviewer pFXH finds the paper incremental. In fact, as reviewer dQdy points out, the narrative of the paper changes after the revision. This is the sign that the clarity of the work is in general lacking.

**Resubmission Of Major Revision:**

The authors may consider submitting a major revision at a later time.